# Three-Dimensional Hydroelasticity of Multi-Connected Modular Offshore Floating Solar Photovoltaic Farm

Zhi Yung Tay

Engineering Cluster, Singapore Institute of Technology, Singapore 138683, Singapore;
zhiyung.tay@singaporetech.edu.sg

**Abstract:** This paper investigates the hydroelastic responses of offshore floating solar photovoltaic farms (OFPVs). OFPVs usually occupy a large sea space in the order of hectares and structural deformation under wave action has to be taken into consideration due to their huge structural length-to-thickness ratio. The flexible deformation of the structure under hydrodynamic loading is termed the hydroelastic response. In the hydroelastic analysis of an OFPV, the diffraction and radiation of waves have to be taken into account to accurately represent the hydrodynamic loadings on the floating platform. In this study, the numerical model is first validated by comparing the eigenvalues and eigenvectors of an OFPV, obtained from the proposed numerical scheme, with their counterparts obtained from an established finite element software. This is followed by an investigation of the hydroelastic responses of various OFPVs designed in varying layout configurations. The various layout configurations are obtained by altering the floating modular units' dimensions as well as the spacing of the OFPVs when deployed adjacent to each other. The optimal configuration that gives the best performance in terms of the overall smallest response, known as compliance, is then suggested. The results suggest that a long-ish OFPV layout has a lower hydroelastic response and that the motion could be further reduced by rearranging the layout arrangement to increase the global flexural stiffness.

**Keywords:** hydroelastic response; offshore floating solar photovoltaic farm; very large floating structure; modular floating structure

## 1. Floating Solar Photovoltaic Farm

The worldwide energy demand is continuously rising, and finding alternative and more sustainable sources of energy is crucial to mitigate the negative environmental impacts associated with fossil fuel-based electricity generation [1–4]. Floating photovoltaic (FPV) systems, which involve installing solar panels supported on a floating platform and deployed on water bodies such as oceans, lakes, reservoirs, and canals, have emerged as an attractive option to overcome land constraints [5–7]. Over the decades, the cumulative installed capacity of floating solar PV farms (FPV) has been increasing year over year, with installations expanding in oceans, lakes, estuaries, and natural basins [8,9]. The advantages of FPV systems include fewer obstacles to block sunlight, convenience in installation, increased energy efficiency, higher power generation efficiency, and reduced water evaporation [10,11]. However, there are challenges related to wind and wave loads as well as corrosion issues, which differ from onshore conditions [6,12], when deploying floating solar systems in marine environments, specifically in the open sea.

The cost of solar panels can vary by location, e.g., countries located in a tropical region receive higher annual solar irradiation as compared to the sub-tropical countries, thereby making solar energy more enticing for the former. Due to the advancement of solar PV technology over the decades, the cost of solar PV panels has dropped drastically [13], where the price of solar panel installation has significantly decreased by 89% over the past decade [14]. This drives the worldwide adoption of solar PV panels to convert solar

irradiation into electricity as an attractive alternative to hydrocarbon. The monocrystalline solar panel, which is the most energy-efficient option, costs approximately from $1 to $1.50 per watt [15], whereas the polycrystalline solar panel, which is less energy-efficient, costs around from $0.90 to $1 per watt [16]. The solar PV panel price is predicted to continue its downward trend from 2023 onwards, as more polysilicon manufacturers come into operation [17].

The steady reduction in the cost of solar PV panels has resulted in an increase in the solar panel footprint globally as an important initiative to reduce the global reliance on hydrocarbon. The International Energy Agency (IEA) reported that solar PV accounted for 4.5% of global electricity generation in 2022, making it the third largest renewable electricity technology, following hydropower and wind [18]. Some of the world's largest FPV systems deployed in lakes and reservoirs can be found in Asia. For example, the FPV system located in Sirindhorn Reservoir in Thailand covers an area equivalent to about 70 soccer fields and has the capacity to generate 45 MW of power [19]. China is also home to three of the largest FPV systems in the world, i.e., Dingzhuang FPV system in Dezhou (320 MW) [20], the FPV system in the Three Gorges (150 MW) [21], and the CECEP FPV system in An hui (70 MW) [22]. In 2021, Singapore unveiled one of the world's largest FPV systems, spanning an area equivalent to 45 football fields [23]. The solar farm located on a reservoir in western Singapore has a 60 megawatt-peak (MWp) capacity and aims to reduce carbon emissions by about 32 kilotonnes annually. Korea has planned to install the Saemangeum floating solar energy project, which will be the biggest operational floating solar power plant in the world, with a capacity of 2.1 gigawatts (GW), when completed in 2024 [24].

With the promising electricity output from FPV panels deployed on water bodies, energy providers have looked into offshore FPV farms (OFPVs) to be deployed in the open sea, as the ocean offers an enticing option with a theoretical global PV capacity of around 4000 gigawatts [25]. Research suggests that floating solar panels at sea perform nearly 13% better on average than land-based installations, with some months generating up to 18% more energy due to lower temperatures and reduced cloud cover [26]. These OFPVs have increased in size over the years, in the order of hectares. Sunseap OFPV, another solar PV farm located in Singapore, has a 5 MWp and is considered one of the largest offshore solar developments in the world [27]. The project occupies a five-hectare footprint and is estimated to produce up to six megawatt-hours (MWh) of energy per year. At the same time, European energy providers such as the Dutch-Norwegian company SolarDuck are working with the German energy company RWE to build a floating solar plant with a raising deck at the North Sea wind farm [28]. Meanwhile, the Norway-based Ocean Sun has developed a floating rig where the solar panels rest on a base which flexes as the waves pass underneath [29].

## 2. Hydroelastic Response

The increase in size of OFPVs means that the fluid–structure interaction (FSI) becomes prominent, especially under the wave action. The flexible deformation of the structure under hydrodynamic loading is termed the hydroelastic response. As OFPVs are constructed further out to sea, they are exposed to greater wave loadings; therefore, OFPVs such as the one by Ocean Sun, where the PV panels are supported on a floating base, allow the structure some flexibility as wave passes underneath. The allowance for some flexibility means that the structure could be constructed with less rigidity. This is important, especially when the structure is larger, as it could significantly reduce the cost of OFPVs.

Conventional solar PV panels are supported by multi-connected modular floating platforms where the structures are simply assumed to be rigid bodies and structure deformations under wave action are neglected. Song et al. [30] studied the dynamic response of a multi-connected floating solar panel system by assuming that the supported structure was a rigid body. A hydrodynamic analysis to study the motion of floating offshore solar farms subjected to regular waves was also conducted by Al-Yacouby et al. [31], who made the same assumption that the structure was a rigid body. Having said that, the hydroelastic

response of OFPVs has been investigated by researchers such as Sree et al. [32], who considered a multi-scale numerical approach to predict the responses of 6000 interconnected floating modular units where the FSI was solved using the arbitrary Langrangian–Eulerian method. As the solution to the fluid motion involves solving the Navier–Stokes equation, the computation time required to solve the FSI is costly. The computational time could be accelerated by modelling the fluid as a potential flow. This has been carried out by Xu and Wellens [33], who considered a large-scale floating PV farm supported by a membrane. They investigated the fully nonlinear FSI, focusing on the performance of the modules under potential flow, which they solved analytically up to the third order. The interconnected floating modular units in their case were modelled using the Euler Bernoulli-von Kármán beam model.

This paper will study the hydroelastic response of multi-connected floating modular units that serve as a platform to support solar panels. The interconnected floating units are modelled using the Kirchhoff–Love thin plate theory [34,35], which better represents an OFPV which has a small depth relative to its length dimension. The linear potential flow theory is used to represent the fluid, where the fluid is assumed to be inviscid, incompressible, and flow irrotational. The vibration mode of the floating platform is obtained from the proposed numerical scheme and is first validated with that obtained from an established finite element model. The hydroelastic response of an OFPV subject to regular waves is then studied. Various configurations with different module dimensions and spacing of the floating solar platform are considered, and the optimal configuration that gives the best performance in terms of the overall minimum response, known as compliance, is then suggested.

## 3. Problem Definition

### 3.1. Description

An OFPV made of universal modular units is considered. Such a structure is classified as a mat-like or pontoon-type very large floating structure (VLFS), with a large surface area, where its area is supported by buoyancy forces. The OFPV is made up of interconnected modular units to form a grid-like layout configuration as shown in Figure 1, and this reduces the global structure rigidity and, thus, the OFPV deforms flexibly under wave action. The grid-like layout configuration of the OFPV shown in Figure 1 has a total length $L_x$ in the $x$ direction, length $L_y$ in the $y$ direction, and depth $h$ with the origin of the global cartesian coordinate system ($x$-$y$-$z$ axes) located at the free surface $z = 0$. The universal modular unit that makes up the OFPV platform has a length $l_x$ or $l_y$ (see Figure 1) and width $b$, where they are connected to form grid-like configuration as shown in Figure 1. Each rectangular grid has the same dimensions due to the uniform solar PV panel's dimensions. A regular wave with an amplitude $A$, wave period $T$, and wavelength $\lambda$ approaches the floating farm at an angle $\theta$, measured anti-clockwise from the negative $x$ axis. The water depth $D$ is assumed to be constant. For clearer visualization, a three-dimensional OFPV arranged in a five by five grid configuration is given in Figure 2.

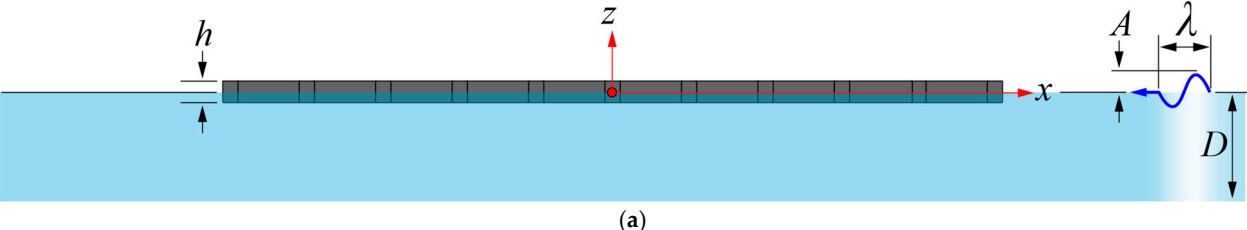

(a)

**Figure 1.** *Cont.*

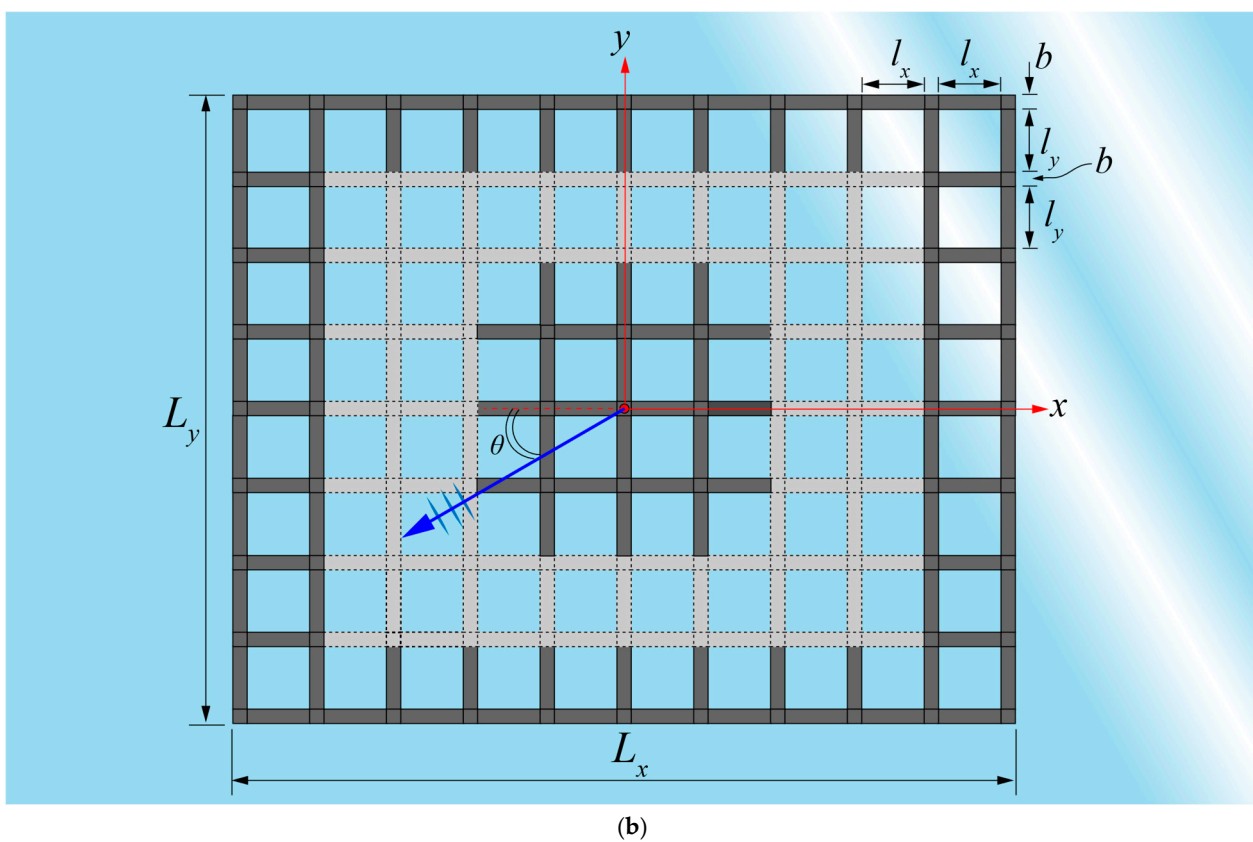

(**b**)

**Figure 1.** Schematic diagram showing (**a**) elevated view and (**b**) plan view of OFPV. Grey dotted modules indicate that the OFPV could be expanded or reduced in size using integrated modular units.

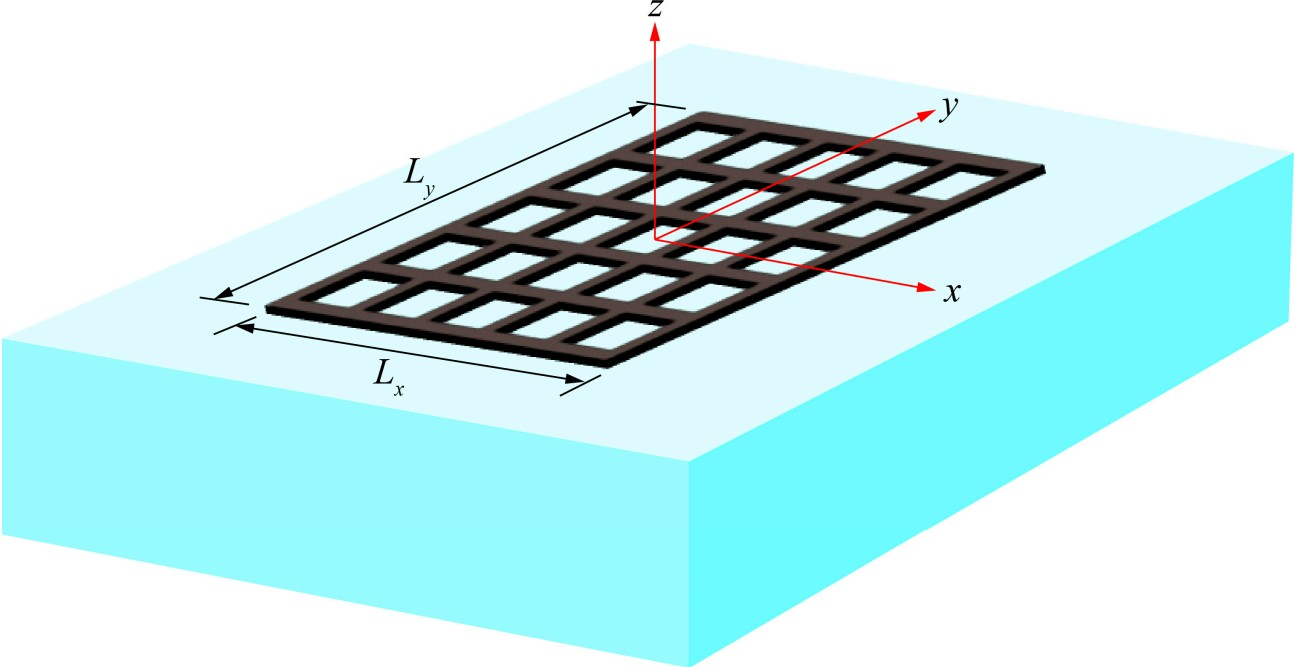

**Figure 2.** Example of OFPV arranged in a 5 × 5 grid configuration. The (number of rows) × (number of columns) for different grid layouts considered in the case studies are given in Figure 3.

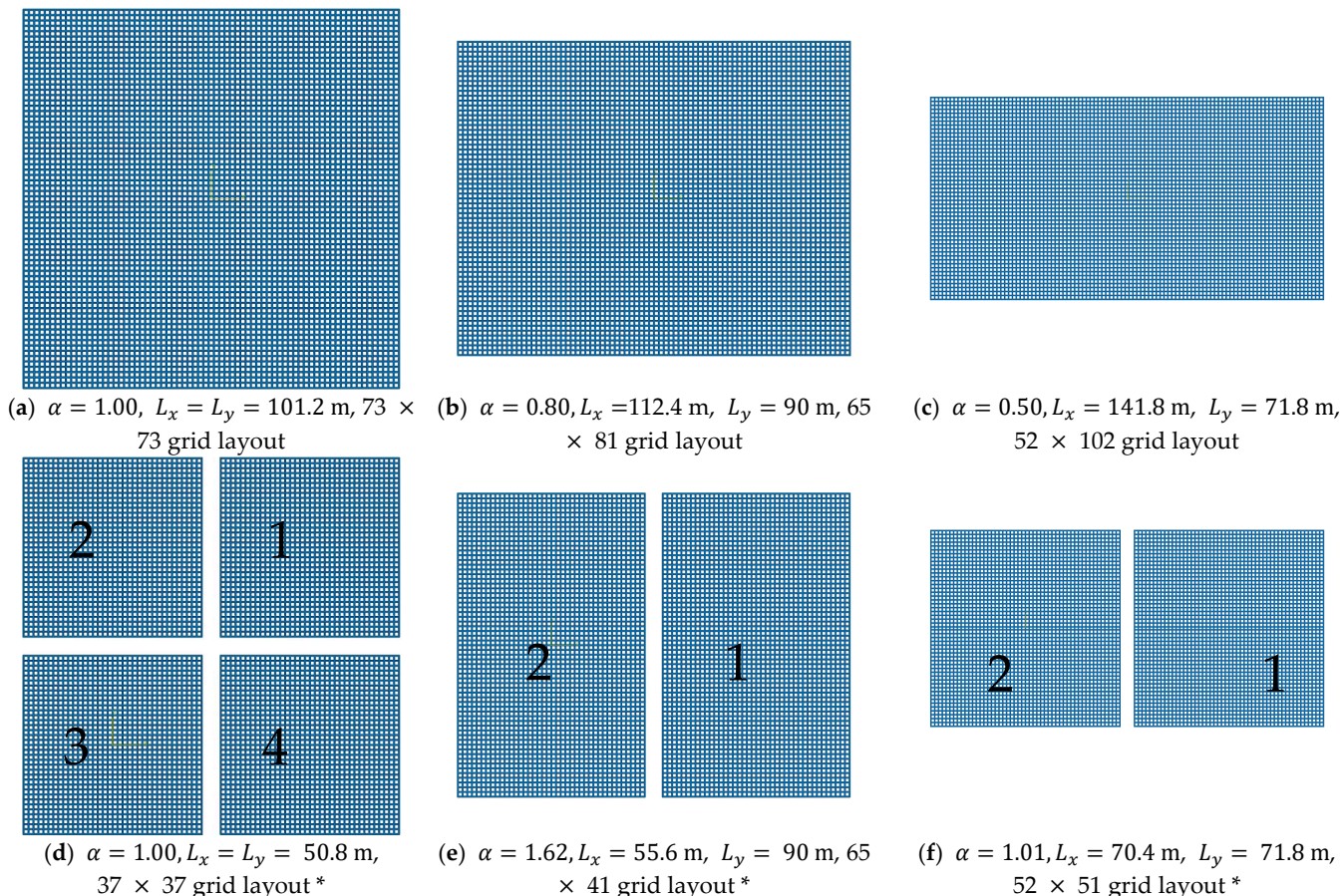

(**a**) $\alpha = 1.00$, $L_x = L_y = 101.2$ m, $73 \times 73$ grid layout

(**b**) $\alpha = 0.80$, $L_x = 112.4$ m, $L_y = 90$ m, $65 \times 81$ grid layout

(**c**) $\alpha = 0.50$, $L_x = 141.8$ m, $L_y = 71.8$ m, $52 \times 102$ grid layout

(**d**) $\alpha = 1.00$, $L_x = L_y = 50.8$ m, $37 \times 37$ grid layout *

(**e**) $\alpha = 1.62$, $L_x = 55.6$ m, $L_y = 90$ m, $65 \times 41$ grid layout *

(**f**) $\alpha = 1.01$, $L_x = 70.4$ m, $L_y = 71.8$ m, $52 \times 51$ grid layout *

**Figure 3.** * means for each OFPV. Layout configurations considered in case studies: (**a**) OFPV-IA: $\alpha = 1.00$, (**b**) OFPV-IB: $\alpha = 0.80$, (**c**) OFPV-IC: $\alpha = 0.50$, (**d**) OFPV-IIA: $\alpha = 1.00$, (**e**) OFPV-IIB: $\alpha = 1.62$, (**f**) OFPV-IIC: $\alpha = 1.01$. Note that $\alpha$ in Figure 2d–f is for each separated OFPV.

*3.2. Case Studies*

Figure 3 shows the six OFPVs arranged in different layout configurations considered in the case studies. Each layout occupies an area of around 10,000 m², equivalent to 1 hectare of sea space. The OFPVs from Figure 3a–c are made up of interconnected floating modular units made of HDPE where each layout differs in the aspect ratio $\alpha$, defined as the ratio of $L_y$ to $L_x$, i.e., $\alpha = L_y/L_x$. Three different $\alpha$, i.e., $\alpha = 1.00$, 0.80 and 0.50, are first considered in Figure 3a, b and c, respectively, for large-scale OFPV connected as one whole piece. Each modular unit has a length $l = l_x = l_y = 1.00$ m, width $b = 0.4$ m and depth $h = 0.2$ m, thereby producing the lengths $L_x$ and $L_y$ as indicated in Figure 3 for the layouts considered.

The layouts in Figure 3a–c are then split into smaller solar farms, with the corresponding layouts shown in Figure 3d–f, respectively, to study the effect of wave interaction between the OFPVs on the reduction in the hydroelastic response. The separated solar farms are placed apart by the spacing $s_p$. For the case studies, three different spacings are considered, i.e., $s_p = 1$ m, 5 m and 10 m.

The consideration of different layouts allows for an investigation of the effect that varying stiffness, expressed as the stiffness coefficient $\beta$, has on the hydroelastic response. The stiffness coefficient is given by $\beta = D/\rho g L^4$, with $D = Eh^3/[12(1 - \nu^2)]$ representing the flexural rigidity, $E$ the Young's modulus, $\nu$ the Poisson ratio, $\rho = 1000$ kg/m³ the water mass density, and $g = 9.81$ m/s² the gravitational acceleration, with the arbitrary length $L$ taken as $L = L_x$. Here, the $E$ is considered as 534 MPa and the mass density of the modular unit is taken as $\rho_p = 960$ kg/m³, adjusted based on the value given in [36]. The principal

dimensions, properties, and parameters considered in the case studies are presented in Table 1.

**Table 1.** Principal dimensions, properties, and parameters considered in case studies.

| | | OFPV Properties | | | | | Regular Waves |
|---|---|---|---|---|---|---|---|
| | | $\alpha$ | $h$ (m) | $L_x \times L_y$ (m $\times$ m) | $T$ (s) | $\lambda$ (m) | $\theta$ (°) |
| OFPV-I | OFPV-IA | 1.00 | | 101.2 × 101.2 | | | |
| | OFPV-IB | 0.800 | | 112.4 × 90.0 | | | |
| | OFPV-IC | 0.500 | | 141.8 × 71.8 | | | |
| OFPV-II † | OFPV-IIA | 1.00 | 0.2 | 50.8 × 50.8 | 3, 4, 5 | 14, 25, 37 | 0, 30, 45, 60, 90 |
| | OFPV-IIB | 1.62 | | 56.4 × 90.0 | | | |
| | OFPV-IIC | 1.01 | | 70.4 × 71.8 | | | |
| OFPV-III | OFPV-IIIA | | 0.600, 0.200, | 6.0 × 6.0 | Note: | | |
| | OFPV-IIIB | 1.00 | 0.060, 0.030, | 20.0 × 20.0 | • For Free Vibration Analysis | | |
| | OFPV-IIIC | | 0.006 | 39.8 × 39.8 | | | |

Aspect ratio for each separated OFPV. † spacing $s_p = 1$ m, 5 m, and 10 m; Young's modulus $E = 534$ MPa, mass density $\rho_p = 960$ kg/m$^3$, and water depth $D = 10$ m.

For simplicity, the OFPV in Figure 3a–c are termed OFPV-IA, OFPV-IB and OFPV-IC, respectively, whereas Figure 3d–f are termed OFPV-IIA, OFPV-IIB and OFPV-IIC, respectively. Note that the (number of rows) × (number of columns) in the grid layout is denoted in the sub-captions in Figure 3. For instance, OFPV-IA has a 73 × 73 grid layout, similar to Figure 2. For this paper, an additional three small-scale floating solar layouts under OFPV-III are considered for the validation of the thin plate theory to be presented in Section 5.1.

## 4. Mathematic Formulation

### 4.1. Water Domain

The water is modelled using the potential theory, assuming that the water is inviscid and incompressible, and its flow is irrotational. Based on these assumptions, the fluid motion may be represented by a velocity potential $\Phi(x, y, z, t)$, and the water is assumed to oscillate in a steady-state harmonic motion with the circular frequency $\omega$. Introducing the variables in the nondimensional form [37]: $\overline{x} = x/L, \overline{y} = y/L, \overline{z} = z/L, \overline{t} = t\sqrt{g/L}$, $\overline{\omega} = \omega\sqrt{L/g}$ and $\overline{\Phi} = \Phi/(L\sqrt{Lg})$, where $L$ is an arbitrarily chosen length parameter and $g$ is the gravitational acceleration.

The velocity potential $\overline{\Phi}(x, y, z, t)$ can be expressed as,

$$\overline{\Phi}(\overline{x}, \overline{y}, \overline{z}, \overline{t}) = Re\left\{\overline{\phi}(\overline{x}, \overline{y}, \overline{z})e^{-i\overline{\omega}\overline{t}}\right\} \tag{1}$$

The single-frequency velocity potential $\overline{\phi}(\overline{x}, \overline{y}, \overline{z})$ must satisfy the Laplace Equation (2) [38] and the boundary conditions on the surfaces as shown in Figure 4.

$$\nabla^2\overline{\phi} = 0 \text{ in } \Omega \tag{2}$$

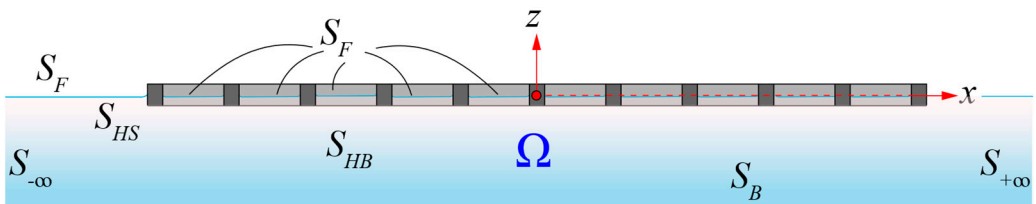

**Figure 4.** Mathematical domain.

These boundary conditions are given as follows [38]:

$$\frac{\partial \overline{\phi}}{\partial z} = i\overline{\omega} \cdot \overline{w} \text{ on } S_{HB} \tag{3}$$

$$\frac{\partial \overline{\phi}}{\partial n} = 0 \quad \text{on } S_{HS} \tag{4}$$

$$\frac{\partial \overline{\phi}}{\partial \overline{z}} = \overline{\omega}^2 \overline{\phi} \quad \text{on } S_F \tag{5}$$

$$\frac{\partial \overline{\phi}}{\partial \overline{z}} = 0 \quad \text{on } S_B \tag{6}$$

where $n$ is the unit normal vector to the surface $S$. The wave velocity potential must also satisfy the Sommerfield radiation condition at the artificial fluid boundary at infinity: $S_{\pm\infty}$ as $(\overline{x}, \overline{y}) \to \infty$ [38].

$$\lim_{|(\overline{x},\overline{y})|\to\pm\infty} \sqrt{|(\overline{x},\overline{y})|} \left( \frac{\partial}{\partial |(\overline{x},\overline{y})|} - i\overline{k} \right) (\overline{\phi} - \overline{\phi}_{In}) = 0 \quad \text{on } S_{\pm\infty} \tag{7}$$

$\overline{k} = 2\pi/\overline{\lambda}$ is the standard nondimensional wave number where $\overline{\lambda} = \lambda/L$ and $\lambda$ is the wavelength. $\overline{\phi}_{In}$ is the incident velocity potential given as:

$$\overline{\phi}_{In} = \frac{A}{\overline{\omega}} \frac{\cosh \overline{k}\overline{z}}{\cosh \left[ \overline{k}\left(\overline{z} + \overline{H}\right) \right]} e^{i\overline{k}(\overline{x}\cos\theta + \overline{y}\cos\theta)} \tag{8}$$

### 4.2. Structure Domain

An OFPV is usually made up of multi-modular units to form a large platform. Such a large platform is classed as a VLFS, for which the hydroelasticity is significant. The OFPV where the length dimensions, i.e., $L_x$ and $L_y$, are significantly larger than the depth $h$ is thus modelled as a solid plate by using the Kirchhoff–Love plate theory [34,35,39]. The solid plate is assumed to be perfectly flat with free edges, i.e., the bending moment and shear force are equal to zero, and the plate material is commonly assumed to be isotropic and obey Hooke's Law [40]. As the OFPV has a wide footprint, its vertical motion is significantly more important compared to the horizontal motion. Therefore, the Kirchhoff–Love plate theory described the plate by using three variables, i.e., the nondimensional deflection normalised with the arbitrary length $L$, i.e., $\overline{w}(\overline{x}, \overline{y})$, the rotation about the $y$ axis $\overline{\theta}_{\overline{x}}(\overline{x}, \overline{y})$, and the rotation about the $x$ axis $\overline{\theta}_{\overline{y}}(\overline{x}, \overline{y})$, as follows [34,35,39]:

$$\beta \left( \frac{\partial^4 \overline{w}}{\partial x^4} + 2\frac{\partial^4 \overline{w}}{\partial x^2 \partial y^2} + \frac{\partial^4 \overline{w}}{\partial y^2} \right) + \overline{\rho}_p \overline{h}\overline{\omega}^2 \overline{w} = \overline{p}(\overline{x}, \overline{y}, \overline{z}) \tag{9}$$

where $\beta = D/\rho g L^4$, $\overline{\rho}_p$ is the specific mass density of the plate (normalised with reference density $\rho = 1000$ kg/m$^3$), $\overline{h}$ is the nondimensional thickness of the plate, $D = Eh^3/[12(1-\nu^2)]$ is the flexural rigidity, $E$ is the Young's modulus, and $\nu$ is the Poisson ratio. The pressure $\overline{p}(\overline{x}, \overline{y}, \overline{z})$ in (9) comprises the hydrostatic and hydrodynamic pressures.

The hydrostatic and hydrodynamic pressures $\overline{p}(\overline{x}, \overline{y}, \overline{z})$ acting on the bottom of the structure (i.e., $z = 0$) are given by the linearised Bernoulli equation:

$$\overline{p}\left(\overline{x}, \overline{y}, -\overline{h}\right) = i\overline{\omega}\overline{\phi}\left(\overline{x}, \overline{y}, -\overline{h}\right) + \overline{w}(\overline{x}, \overline{y}) \tag{10}$$

### 4.3. Decoupling of Governing Equations Using Modal Expansion Method

The plate-governing Equation (9) indicates that the response of the plate $\overline{w}$ is coupled with the fluid motions (or velocity potential $\overline{\phi}$). On the other hand, the fluid motion can only be obtained when the plate deflection $\overline{w}$ is specified in the boundary condition (3). To decouple this interaction problem into a hydrodynamic problem, in terms of the velocity potential, and a plate vibration problem, in terms of the generalised displacement, we adopt the modal expansion method as proposed by Newman [41]. In this method, the deflection of the plate $\overline{w}$ is expanded by a series of the products of the modal functions $\mathcal{Z}_l$ and their corresponding complex amplitudes $\zeta_j(\overline{x}, \overline{y})$ [42]:

$$\overline{w}(\overline{x}, \overline{y}) = \sum_{l=1}^{N_m} \mathcal{Z}_l(\overline{x}, \overline{y}) \cdot \zeta_l(\overline{x}, \overline{y}) \tag{11}$$

where $N_m$ denotes the total number of modes taken in the plate analysis.

As the problem is linear, the total velocity potential can be represented by a linear superposition of the diffracted part $\overline{\phi}_D$ on the radiated part $\overline{\phi}_R$. By using the modal expansion method, the total velocity potential $\overline{\varphi}(x, y, z)$ may be expressed as [43]:

$$\overline{\phi}(\overline{x}, \overline{y}, \overline{z}) = \overline{\phi}_D(\overline{x}, \overline{y}, \overline{z}) + \overline{\phi}_R(\overline{x}, \overline{y}, \overline{z}) = \overline{\phi}_D(\overline{x}, \overline{y}, \overline{z}) + i\overline{\omega} \sum_{l=1}^{N_m} \mathcal{Z}_l(\overline{x}, \overline{y}) \cdot \overline{\phi}_l(\overline{x}, \overline{y}) \tag{12}$$

where $\overline{\phi}_D$ is computed from the sum of the incident wave $\overline{\phi}_{In}$ and scattered wave $\overline{\phi}_S$:

$$\frac{\partial \overline{\phi}_D}{\partial n} = \frac{\partial \overline{\phi}_{In}}{\partial n} + \frac{\partial \overline{\phi}_S}{\partial n} = 0 \tag{13}$$

Here, $\overline{\phi}_{l=1,2,\ldots,N_m}$ is the radiation potential corresponding to the unit-amplitude motion of the $l$-th modal function. Note that the complex amplitudes $\mathcal{Z}_l$ in (12) are assumed to be the same values as $\mathcal{Z}_l$ in (11) [41].

By substituting (11) and (12) into the Laplace Equation (2) and the fluid boundary conditions (3) into the Sommerfeld radiation condition (7), we arrive at the following decoupled governing equation and boundary conditions for each of the unit-amplitude radiation potentials (i.e., for $l = 1, 2, \ldots, N_m$) and the diffraction potential (i.e., for $l = D$).

$$\nabla^2 \overline{\phi}_l = 0, \quad \text{in } \Omega \tag{14}$$

$$\frac{\partial \overline{\phi}_l}{\partial \overline{z}} = 0, \quad \text{on } S_B \tag{15}$$

$$\frac{\partial \overline{\phi}_l}{\partial \overline{z}} = \begin{cases} i\omega \mathcal{Z}_l & \text{for } l = 1, 2, \ldots, N_m \\ 0 & \text{for } l = D \end{cases}, \quad \text{on } S_{HB} \tag{16}$$

$$\frac{\partial \overline{\phi}_l}{\partial \overline{z}} = 0, \quad \text{on } S_{HS} \tag{17}$$

$$\frac{\partial \overline{\phi}_l}{\partial \overline{z}} = -\overline{\omega}^2 \varphi_l, \quad \text{on } S_{HB} \tag{18}$$

$$\lim_{|(\overline{x}, \overline{y})| \to \infty} \sqrt{|(\overline{x}, \overline{y})|} \left( \frac{\partial}{\partial |(\overline{x}, \overline{y})|} - i\overline{k} \right) (\overline{\phi}_l - \overline{\phi}_{In}) = 0, \quad \text{on } S_{\pm\infty} \tag{19}$$

The boundary value problem for the diffracted potential and each of the unit-amplitude radiation potentials is defined by (15) to (19). This boundary value problem could be solved by using the boundary element method whereas the water-plate equation (9) can be solved by using the finite element method once the velocity potential is obtained.

### 4.4. BEM for Solving Boundary Integral Equation

The Laplace Equation (14), together with the boundary conditions (15)–(19), on the surface *S* are transformed into a boundary integral equation (BIE) by using Green's Second Theorem. This is executed via a free surface Green's function that satisfies the surface boundary condition at the free water surface $S_F$, at the seabed $S_B$, and at the infinity $S_\infty$. Hence, only the wetted surface of the bodies $S_H \ni (S_{HB} \cup S_{HS})$ needs to be discretised into panels so that the boundary element method can be used to solve for the diffracted and radiated potential. The boundary integral equation can be written as [44]:

$$\overline{\phi}_k(\overline{\mathbf{x}}) = \int_{S_H} \left[ iG(\overline{\mathbf{x}}, \overline{\boldsymbol{\xi}}) \cdot \frac{\partial \overline{\phi_k}(\mathbf{x})}{\partial n} - \frac{\partial G(\overline{\mathbf{x}}, \overline{\boldsymbol{\xi}})}{\partial n} \cdot \overline{\phi}_k(\overline{\mathbf{x}}) \right] \cdot dS_H \quad \text{for } k = R \text{ or } D \tag{20}$$

where $G(\overline{\mathbf{x}}, \overline{\boldsymbol{\xi}})$ is the free surface Green's function given in [42], with $\overline{\mathbf{x}} = (\overline{x}, \overline{y})$ representing the field points and $\overline{\boldsymbol{\xi}} = (\overline{\xi}, \overline{\eta})$ representing the source points. The evaluation of $G(\overline{\mathbf{x}}, \overline{\boldsymbol{\xi}})$ requires the distribution of source point $\overline{\boldsymbol{\xi}}$ at all of the panels and evaluating its influence on the field point $\overline{\mathbf{x}}$.

By imposing the boundary condition (13) and expressing the plate deflection using the modal expansion method (11), Equation (20) can be written as:

$$\overline{\phi}_k(\overline{\mathbf{x}}) = \left[ 2\pi + \int_{S_H} \left( \frac{\partial G(\overline{\mathbf{x}}, \overline{\boldsymbol{\xi}})}{\partial n} \right) \cdot dS_H \right]^{-1} \times \begin{cases} i\overline{\omega} \int_{S_H} G(\overline{\mathbf{x}}, \overline{\boldsymbol{\xi}}) \cdot \mathcal{Z}_l \cdot \zeta_l \cdot dS_H & \text{for } k = R \\ 4\pi \overline{\phi}_{In}(\overline{\mathbf{x}}) & \text{for } k = D \end{cases} \tag{21}$$

Assuming that the floating body is discretized into $N_e$ number of elements, also called panels, Equation (21) can be written for $\overline{\phi}_l$ and $\overline{\phi}_D$ separately in the form:

$$\overline{\phi}_l(\overline{\mathbf{x}}) = \frac{1}{2\pi} \left[ 1 + \frac{1}{2\pi} \int_{S_H} \left( \frac{\partial G(\overline{\mathbf{x}}, \overline{\boldsymbol{\xi}})}{\partial n} \right) \cdot dS_H \right]^{-1} \times \int_{S_H} G(\overline{\mathbf{x}}, \overline{\boldsymbol{\xi}}) \cdot \mathcal{Z}_l \cdot dS_H \tag{22}$$

$$\overline{\phi}_D(\overline{\mathbf{x}}) = 2 \left[ 1 + \frac{1}{2\pi} \int_{S_H} \left( \frac{\partial G(\overline{\mathbf{x}}, \overline{\boldsymbol{\xi}})}{\partial n} \right) \cdot dS_H \right]^{-1} \times \overline{\phi}_{In}(\overline{\mathbf{x}}) \tag{23}$$

where $\mathcal{Z}_l$ in (22) is the mode shapes (eigenvectors) obtained by performing a free vibration analysis on the plate.

The nondimensional added mass $\overline{\mathcal{A}}$ and radiated damping $\overline{\mathcal{B}}$ are related to $\overline{\phi}_R$ with the following relationship [45]:

$$\overline{\mathcal{A}} - \frac{1}{\overline{\omega}} \overline{\mathcal{B}} = \int_{S_H} n \cdot \overline{\phi}_l dS \tag{24}$$

### 4.5. Equation of Motion for Water-Plate Model

By integrating (9) with respect to the hull-wetted surface to obtain the forces, the decoupled equation of motion of the water-structure problem can be obtained as:

$$\left[ \overline{\omega}^2 \overline{M} + \overline{K} \right] \cdot \overline{W} = \overline{F} \tag{25}$$

where $\overline{W} = (\overline{w}, \overline{\theta}_x, \overline{\theta}_y)$. As only the motion in the vertical direction is considered, Equation (9) needs to be multiplied by the vertical wetted surface ($S_{HB}$) of the OFPV, i.e., its waterplane area $\overline{A}_{wp}$ to obtain the forces. Therefore, in (25), the nondimensional flexural stiffness $K$ and mass $M$ are obtained from:

$$\overline{K} = \beta \cdot \int_{S_H} \left( \frac{\partial^4 \overline{w}}{\partial x^4} + 2 \frac{\partial^4 \overline{w}}{\partial x^2 \partial y^2} + \frac{\partial^4 \overline{w}}{\partial y^2} \right) \cdot dS_H \tag{26}$$

$$\overline{M} = \overline{\rho}_p \cdot \int_{S_H} \overline{h} \cdot dS_H \tag{27}$$

The force $\overline{F}$ is derived from the pressure $\overline{p}$ as:

$$\overline{F} = i\overline{\omega}\overline{\phi} + \int_{S_{HB}} \overline{w} \cdot dS_{HB} \tag{28}$$

The second term in (28) contributes to the hydrostatic force $\overline{F}_{hs}$:

$$\overline{F}_{hs} = \int_{S_{HB}} \overline{w} \cdot dS_{HB} = \overline{A}_{wp} \cdot \overline{w} \tag{29}$$

On the other hand, the first term in (28) is the hydrodynamic force $\overline{F}_{hd}$ derived from the velocity potential $\overline{\phi}$. According to (24), the radiated potential $\overline{\phi}_R$ contributes to the added mass $\overline{\mathcal{A}}$ and radiated damping $\overline{\mathcal{B}}$. The diffracted potential $\overline{\phi}_D$ contributes to the exciting force, i.e.:

$$\overline{F}_e = i\overline{\omega} \cdot \overline{\phi}_D \tag{30}$$

Therefore, we have the equation of motion (25) written as:

$$\left[ \overline{\omega}^2 (\overline{M} + \overline{\mathcal{A}}) - i\overline{\omega}\overline{\mathcal{B}} + (\overline{K} + I) \right] \cdot \overline{W} = \overline{F}_e \tag{31}$$

where $I = \mathrm{diag}\begin{bmatrix} \overline{A}_{wp} & 0 & 0 \end{bmatrix}$.

## 5. Model Validation

### 5.1. Validation of Thin Plate Model

The OFPV modelled by using the thin plate theory is first validated with its counterparts obtained from the finite element software ABAQUS. Here, three different configurations of OFPVs are considered, namely OFPV-IIIA, OFPV-IIIB, and OFPV-IIIC, with their details summarized in Table 1. The three layout configurations are presented in Figure 5. The lengths of the $x$ and $y$ axes are taken to be the same, i.e., $\alpha = 1.00$, so that the layout configuration has a square waterplane.

The natural frequencies are obtained by performing the free vibration analysis of the OFPV with free edge boundary conditions. Note that the free edge boundary conditions require the bending moments, twisting moments, and shear forces to vanish at the free edges of the OFPV. The free vibration analysis is executed by solving the eigenvalue problem, where the eigenvalues correspond to the nondimensional natural wave frequencies $\overline{\omega}_n$ and the eigenvectors $\mathcal{Z}$ to the vibration modes, given as:

$$\left[ \overline{K} - \overline{\omega}_n^2 \overline{M} \right] \cdot \mathcal{Z} = 0 \tag{32}$$

The accurate prediction of the eigenvalues and eigenvectors is important to accurately represent the floating plate using the modal expansion method, as explained in Section 4.3. The modal expansion method could reduce the matrix sizes involved in solving the fluid–structure coupled Equation (25).

Five different module depths are considered as presented in Table 1, i.e., $h = 0.600$ m, $0.200$ m, $0.060$ m, $0.030$ m, and $0.006$ m, to study the convergence of the present numerical model for the thin plate theory. The comparisons of the natural frequencies $f_n$ of OFPV-IIIA, OFPV-IIIB, and OFPV-IIIC are presented in Table 2, Table 3, and Table 4, respectively, where $f_n$ is given by:

$$f_n = \sqrt{\left( \frac{\overline{\omega}_n}{2\pi} \right)^2 \cdot \frac{g}{L}}, \quad \text{unit in Hertz(hz).} \tag{33}$$

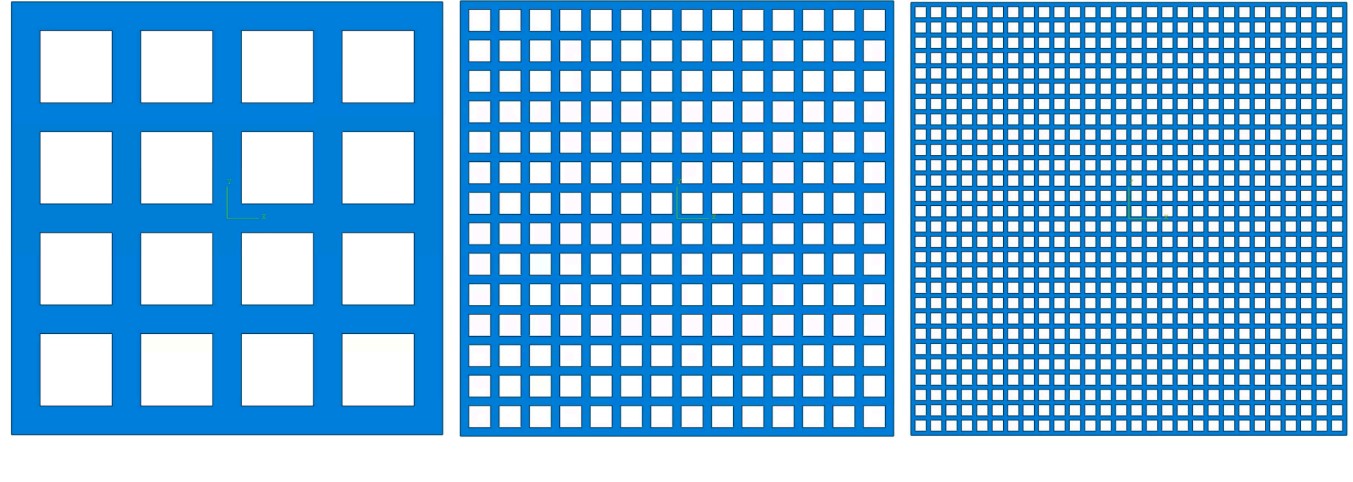

(**a**) $L_x = L_y = 6.0$ m          (**b**) $L_x = L_y = 20.0$ m          (**c**) $L_x = L_y = 39.8$ m

**Figure 5.** Layout considered for validation of finite element model. (**a**) OFPV-IIIA, (**b**) OFPV-IIIB, (**c**) OFPV-IIIC.

**Table 2.** Comparison of natural frequency between present model with ABAQUS for OFPV-IIIA ($L = 6$ m, $\Delta_e = 0.1$ m).

| $h$ (m) | 0.6000 | | 0.2000 | | 0.0600 | | 0.0300 | | 0.0060 | |
|---|---|---|---|---|---|---|---|---|---|---|
| $\overline{h}$ | 0.1000 | | 0.0333 | | 0.0100 | | 0.0050 | | 0.0010 | |
| Mode | $f_n$(hz) | | $f_n$(hz) | | $f_n$(hz) | | $f_n$(hz) | | $f_n$(hz) | |
| | PM | ABQ | PM | ABQ | PM | ABQ | PM | ABQ | PM | ABQ |
| 1 (RBM *) | 0.0000 | 0.0000 | 0.0000 | 0.0000 | 0.0000 | 0.0000 | 0.0000 | 0.0000 | 0.0000 | 0.0000 |
| 2 (RBM *) | 0.0000 | 0.0000 | 0.0000 | 0.0000 | 0.0000 | 0.0000 | 0.0000 | 0.0000 | 0.0000 | 0.0000 |
| 3 (RBM *) | 0.0000 | 0.0000 | 0.0000 | 0.0000 | 0.0000 | 0.0000 | 0.0000 | 0.0000 | 0.0000 | 0.0000 |
| 4 | 7.7666 | 4.8689 | 2.5889 | 2.2938 | 0.7767 | 0.7565 | 0.3883 | 0.3848 | 0.0777 | 0.0776 |
| 5 | 9.4832 | 8.8503 | 3.1611 | 3.0668 | 0.9483 | 0.9391 | 0.4742 | 0.4719 | 0.0948 | 0.0947 |
| 6 | 9.6313 | 9.1389 | 3.2104 | 3.1534 | 0.9631 | 0.9567 | 0.4816 | 0.4795 | 0.0963 | 0.0961 |
| 7 | 18.3765 | 13.3460 | 6.1255 | 5.5782 | 1.8377 | 1.7991 | 0.9188 | 0.9118 | 0.1838 | 0.1836 |
| 8 | 18.3765 | 13.3520 | 6.1255 | 5.5789 | 1.8377 | 1.7993 | 0.9188 | 0.9119 | 0.1838 | 0.1836 |
| 9 | 26.5784 | 13.8560 | 8.8595 | 8.5653 | 2.6578 | 2.6331 | 1.3289 | 1.3229 | 0.2658 | 0.2653 |
| 10 | 26.5784 | 20.8870 | 8.8595 | 8.5663 | 2.6578 | 2.6334 | 1.3289 | 1.3231 | 0.2658 | 0.2654 |
| 11 | 33.1127 | 20.9550 | 11.0376 | 9.9532 | 3.3113 | 3.2373 | 1.6556 | 1.6426 | 0.3311 | 0.3310 |

* RBM: Rigid body motion, PM: Present Method, ABQ: ABAQUS.

The depth $h$ is normalized with the length $L$ in this study, where the nondimensional depth is defined as $\overline{h} = h/L$.

The comparison of the natural frequencies $f_n$ for OFPV-IIIA in Table 2 shows that the accuracy of the present numerical model is inaccurate when $\overline{h} > 0.0100$, i.e., $\overline{h} = 0.0333$ and $\overline{h} = 0.1000$. When $\overline{h} \leq 0.0100$, the $f_n$ predicted by the present model begins to agree with its counterparts obtained from ABAQUS, and the difference between the two models reduces when $\overline{h}$ becomes smaller. Similar observations can be found for OFPV-IIIB and OFPV-IIIC, in Tables 3 and 4, respectively. With the increase in the length $L$ of the OFPV, there is an improvement in the accuracy of the present numerical model in predicting the $f_n$ of the solar farm with a thicker depth. For example, for OFPV-IIIA, in Table 2, the $f_n$ predicted by the present numerical model is acceptable for $h \leq 0.060$ m but for OFPV-IIIB, the $f_n$ predicted by the present model is in good agreement with its counterparts from ABAQUS up to $h = 0.2000$ m. Therefore, this implies that the present thin plate model is accurate for a small plate thickness-to-length ratio, i.e., $\overline{h}$. As the OFPVs considered in OFPV-I and



OFPV-II have a small $\bar{h}$, the present model can thus be used for hydroelasticity modelling. The free vibration analysis for OFPV-I is given in the next section.

**Table 3.** Comparison of natural frequency between present model with ABAQUS for OFPV-IIIB ($L = 20$ m, $\Delta_e = 0.2$ m).

| $h$ | 0.600 | | 0.200 | | 0.060 | | 0.030 | | 0.006 | |
| $\bar{h}$ | 0.0300 | | 0.0100 | | 0.0030 | | 0.0015 | | 0.0003 | |
| Mode | $f_n$(hz) | | $f_n$(hz) | | $f_n$(hz) | | $f_n$(hz) | | $f_n$(hz) | |
| | PM | ABQ | PM | ABQ | PM | ABQ | PM | ABQ | PM | ABQ |
|---|---|---|---|---|---|---|---|---|---|---|
| 1 (RBM *) | 0.0000 | 0.0000 | 0.0000 | 0.0000 | 0.0000 | 0.0000 | 0.0000 | 0.0000 | 0.0000 | 0.0000 |
| 2 (RBM *) | 0.0000 | 0.0000 | 0.0000 | 0.0000 | 0.0000 | 0.0000 | 0.0000 | 0.0000 | 0.0000 | 0.0000 |
| 3 (RBM *) | 0.0000 | 0.0000 | 0.0000 | 0.0000 | 0.0000 | 0.0000 | 0.0000 | 0.0000 | 0.0000 | 0.0000 |
| 4 | 0.7519 | 0.4813 | 0.2506 | 0.2230 | 0.0752 | 0.0732 | 0.0376 | 0.0372 | 0.0075 | 0.0075 |
| 5 | 0.9070 | 0.8686 | 0.3023 | 0.2946 | 0.0907 | 0.0896 | 0.0454 | 0.0449 | 0.0091 | 0.0090 |
| 6 | 0.9123 | 0.8803 | 0.3041 | 0.2960 | 0.0912 | 0.0901 | 0.0456 | 0.0453 | 0.0091 | 0.0091 |
| 7 | 1.7621 | 1.3053 | 0.5874 | 0.5360 | 0.1762 | 0.1722 | 0.0881 | 0.0873 | 0.0176 | 0.0176 |
| 8 | 1.7621 | 1.3054 | 0.5874 | 0.5360 | 0.1762 | 0.1722 | 0.0881 | 0.0873 | 0.0176 | 0.0176 |
| 9 | 2.5090 | 2.2959 | 0.8363 | 0.8136 | 0.2509 | 0.2477 | 0.1254 | 0.1243 | 0.0251 | 0.0249 |
| 10 | 2.5090 | 2.3981 | 0.8363 | 0.8137 | 0.2509 | 0.2477 | 0.1254 | 0.1243 | 0.0251 | 0.0249 |
| 11 | 3.2480 | 2.3982 | 1.0827 | 0.9803 | 0.3248 | 0.3171 | 0.1624 | 0.1608 | 0.0325 | 0.0324 |

\* RBM: Rigid body motion, PM: Present Method, ABQ: ABAQUS.

**Table 4.** Comparison of natural frequency between present model with ABAQUS for OFPV-IIIC ($L = 39.8$ m, $\Delta_e = 0.2$ m).

| | 0.600 | | 0.200 | | 0.060 | | 0.030 | | 0.006 | |
| $\bar{h}$ | 0.0146 | | 0.0050 | | $1.5075\times10^{-3}$ | | $7.5377\times10^{-4}$ | | $1.5075\times10^{-4}$ | |
| Mode | $f_n$(hz) | | $f_n$(hz) | | $f_n$(hz) | | $f_n$(hz) | | $f_n$(hz) | |
| | PM | ABQ | PM | ABQ | PM | ABQ | PM | ABQ | PM | ABQ |
|---|---|---|---|---|---|---|---|---|---|---|
| 1 (RBM *) | 0.0000 | 0.0000 | 0.0000 | 0.0000 | 0.0000 | 0.0000 | 0.0000 | 0.0000 | 0.0000 | 0.0000 |
| 2 (RBM *) | 0.0000 | 0.0000 | 0.0000 | 0.0000 | 0.0000 | 0.0000 | 0.0000 | 0.0000 | 0.0000 | 0.0000 |
| 3 (RBM *) | 0.0000 | 0.0000 | 0.0000 | 0.0000 | 0.0000 | 0.0000 | 0.0000 | 0.0000 | 0.0000 | 0.0000 |
| 4 | 0.0195 | 0.1255 | 0.0651 | 0.0592 | 0.0195 | 0.0190 | 0.0098 | 0.0097 | 0.0020 | 0.0019 |
| 5 | 0.0234 | 0.2251 | 0.0781 | 0.0756 | 0.0234 | 0.0231 | 0.0117 | 0.0116 | 0.0023 | 0.0023 |
| 6 | 0.0236 | 0.2277 | 0.0787 | 0.0760 | 0.0236 | 0.0233 | 0.0118 | 0.0117 | 0.0024 | 0.0024 |
| 7 | 0.0456 | 0.3385 | 0.1521 | 0.1407 | 0.0456 | 0.0446 | 0.0228 | 0.0226 | 0.0046 | 0.0045 |
| 8 | 0.0456 | 0.3385 | 0.1521 | 0.1407 | 0.0456 | 0.0446 | 0.0228 | 0.0226 | 0.0046 | 0.0045 |
| 9 | 0.0649 | 0.5978 | 0.2162 | 0.2090 | 0.0649 | 0.0640 | 0.0324 | 0.0321 | 0.0065 | 0.0064 |
| 10 | 0.0649 | 0.6234 | 0.2162 | 0.2090 | 0.0649 | 0.0640 | 0.0324 | 0.0321 | 0.0065 | 0.0064 |
| 11 | 0.0842 | 0.6234 | 0.2808 | 0.2586 | 0.0842 | 0.0823 | 0.0421 | 0.0417 | 0.0084 | 0.0084 |

\* RBM: Rigid body motion, PM: Present Method, ABQ: ABAQUS.

*5.2. Vibration Modes and Natural Frequencies of OFPV-I*

The natural frequencies of OFPV-I obtained from the present method and ABAQUS are presented in Table 5. Similar to the previous section, the natural frequencies and vibration modes are obtained by solving the eigenvalue problems (32). The natural frequencies obtained from the present method are found to be in good agreement with their counterparts predicted by ABAQUS. The comparisons of the vibration modes between the present method and ABAQUS for OFPV-IA, OFPV-IB, and OFPV-IC are presented in Figure 6. The results show that the vibration modes obtained from these two models are similar to each other, thus further validating the accuracy of the present model.

**Table 5.** Comparison of natural frequency between present model with ABAQUS for OFPV-I ($h = 0.2000$ m, $\Delta_e = 0.2$ m).

| | OFPV-IA | | OFPV-IB | | OFPV-IC | |
|---|---|---|---|---|---|---|
| $L$ | 101.2 m | | 112.5 m | | 141.8 m | |
| $\overline{h}=h/L$ | $1.9763\times10^{-3}$ | | $1.7778\times10^{-3}$ | | $1.4104\times10^{-3}$ | |
| Mode | $f_n$(hz) | | $f_n$(hz) | | $f_n$(hz) | |
| | PM | ABQ | PM | ABQ | PM | ABQ |
| 1 (RBM *) | 0.0000 | 0.0000 | 0.0000 | 0.0000 | 0.0000 | 0.0000 |
| 2 (RBM *) | 0.0000 | 0.0000 | 0.0000 | 0.0000 | 0.0000 | 0.0000 |
| 3 (RBM *) | 0.0000 | 0.0000 | 0.0000 | 0.0000 | 0.0000 | 0.0000 |
| 4 | 0.0101 | 0.0094 | 0.0098 | 0.0093 | 0.0062 | 0.0060 |
| 5 | 0.0120 | 0.0120 | 0.0102 | 0.0095 | 0.0101 | 0.0092 |
| 6 | 0.0122 | 0.0121 | 0.0153 | 0.0148 | 0.0171 | 0.0165 |
| 7 | 0.0232 | 0.0224 | 0.0226 | 0.0209 | 0.0211 | 0.0193 |
| 8 | 0.0235 | 0.0224 | 0.0255 | 0.0237 | 0.0240 | 0.0232 |
| 9 | 0.0334 | 0.0332 | 0.0271 | 0.0262 | 0.0314 | 0.0296 |
| 10 | 0.0334 | 0.0332 | 0.0407 | 0.0381 | 0.0335 | 0.0320 |
| 11 | 0.0434 | 0.0411 | 0.0422 | 0.0408 | 0.0346 | 0.0323 |

* RBM: Rigid body motion, PM: Present Method, ABQ: ABAQUS.

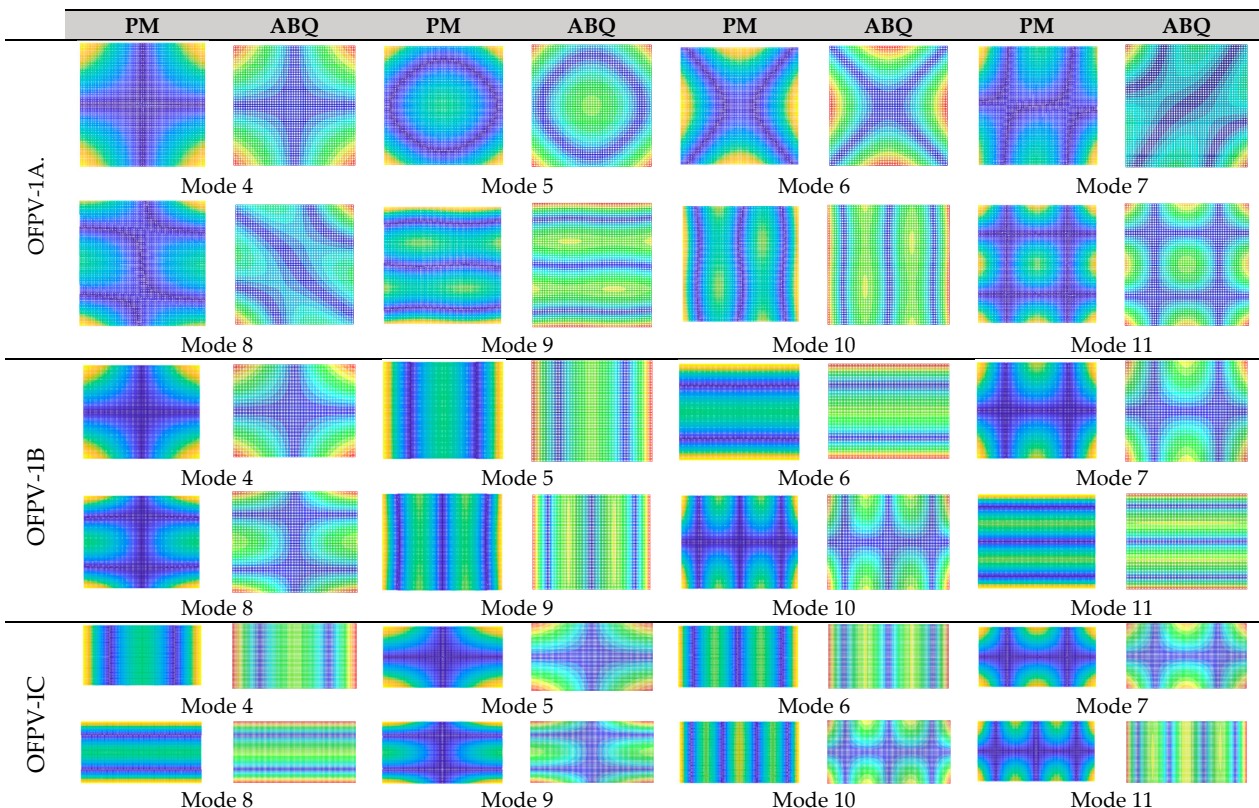

**Figure 6.** Vibration model for OFPV-IA, OFPV-IB, and OFPV-IC.

## 6. Results and Discussions

*6.1. OFPV-I*

6.1.1. Effect of Wave Direction

The hydroelastic response under regular wave conditions predicted by the present numerical method is presented in Figures 7–9 for OFPV-IA, OFPV-IB, and OFPV-IC, respectively. The wave periods considered are in a range from $T = 3$ s to 5 s, similar to the typical wave periods found in tropical regions such as Singapore. The hydroelastic code was

developed in MATLAB® and executed on the Intel® Xeon® Platinum 8180CPU@2.50GHz. A typical hydroelastic response for OFPV-I would take around 18 h to complete. In general, the results show that the hydroelastic response of the OFPV-I is greater under longer wavelengths, i.e., a larger wave period, and reduces with a decrease in wavelength, i.e., a small wave period. However, the elastic deformation of the OFPV-I is observed to be greater when the wavelength-to-structural length ratio is small. Also, it can be seen that the hydroelastic response of OFPV-I is generally higher when subjected to headsea conditions, i.e., $\theta = 0°$. Figures 7–9 how that the hydroelastic response of OFPV-I is the highest, at the forefront of the OFPVs, where incident waves first impact it. The hydroelastic response of the OFPV starts to reduce towards the end of the structure due to the forefront of the OFPV absorbing some of the wave energy. Therefore, it is recommended that the forefronts of OFPVs be made of a structure with higher rigidity that acts as a wave attenuation device to mitigate the hydroelastic response of the structure. Comparing OFPV-I with the reducing aspect ratio $\alpha$ in Figures 8 and 9, where the structures become long-ish (as compared to the squar-ish layout considered in Figure 7), the maximum hydroelastic response of the layout is found to reduce with the decrease in $\alpha$, in which the hydroelastic response for OFPV-IC is the smallest, as shown in Figure 9. The elastic deformation is, however, greater for the long-ish structure due to the reduced flexural rigidity in the longitudinal direction ($x$ axis) when the layout becomes long-ish.

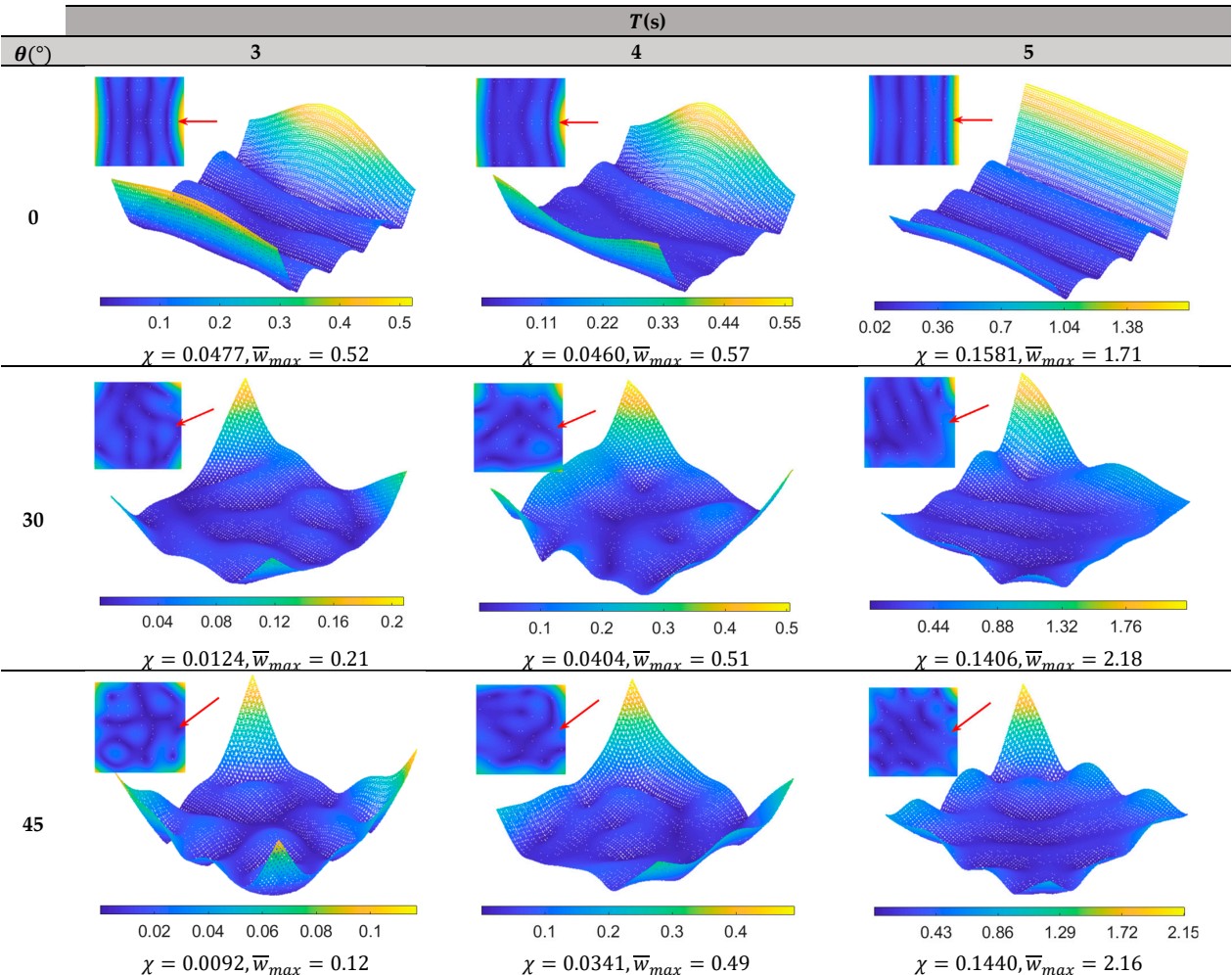

**Figure 7.** Hydroelastic response of OFPV-IA under various wave periods and wave directions. Note: Wave direction is indicated by the red arrow. Small contour figure represents plan view and large contour figure represents isometric view of OFPV-IA.

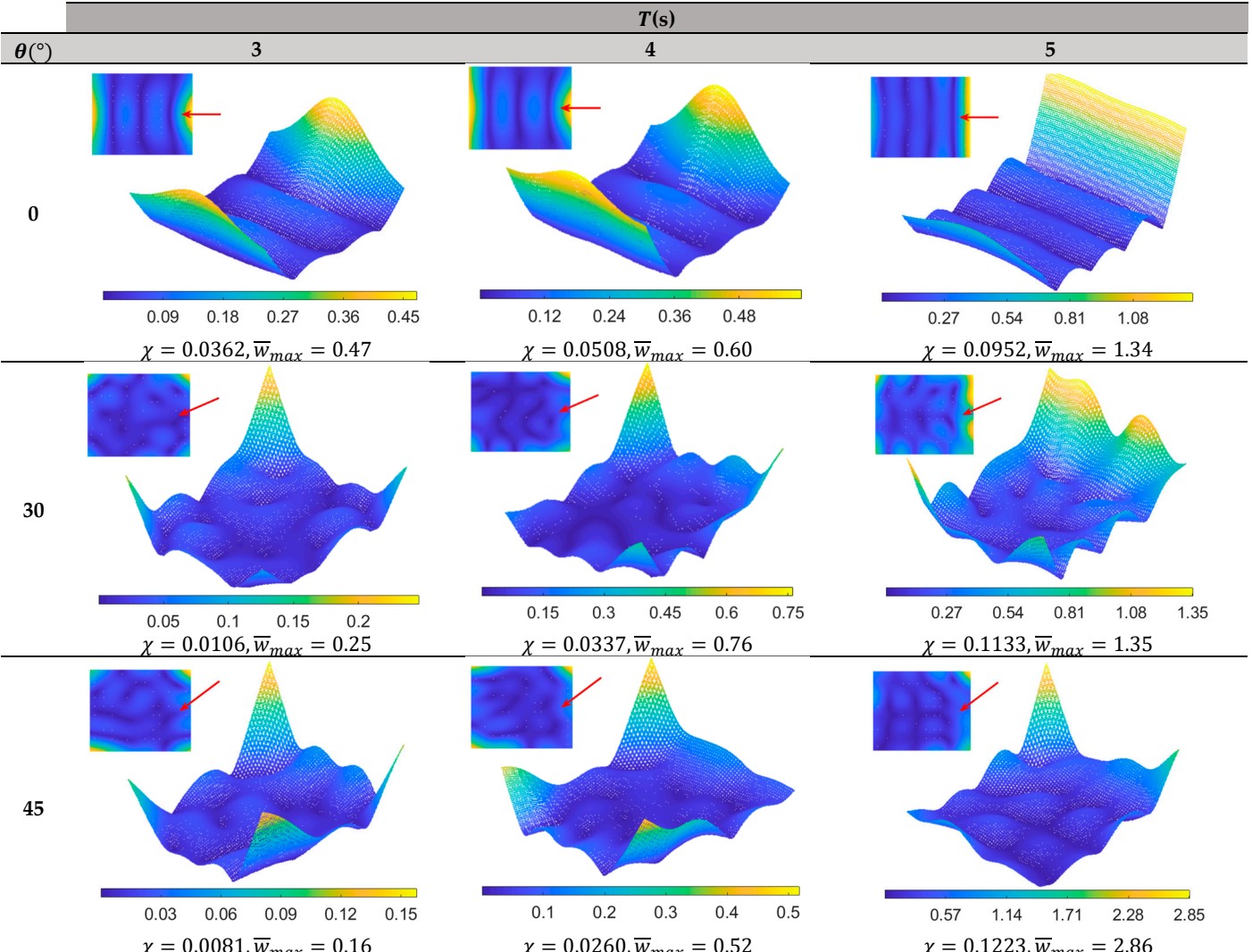

**Figure 8.** Hydroelastic response of OFPV-IB under various wave periods and wave directions. Wave direction is indicated by the red arrow. Small contour figure represents plan view and large contour figure represents isometric view of OFPV-IB.

As compared with the square OFPV-IA floating farm, the hydroelastic response of the long-ish OFPV-IC farm is found to dampen faster, thereby resulting in lower hydroelastic responses. This can be seen clearly in Figure 10, where the elevated view of the hydroelastic response along the centreline of OFPV-IA, OFPV-IB, and OFPV-IC under headsea conditions is presented. Although there is greater elastic deformation in the long-ish OFPVs, i.e., OFPV-IB and OFPV-IC, the magnitude of the elastic deformation is smaller as compared to the square OFPV-IA, thus resulting in an overall smaller response for the former. By comparing the compliance $\chi$ for OFPV-IA, OFPV-IB, and OFPV-IC given in Figures 7 and 8, respectively, there is a decrease in $\chi$ by 26.8%, 29.6%, and 30.5% when compared with OFPV-IA for $T = 3$ s, 4 s, and 5 s, respectively, whereas the decrease in $\chi$ when compared to OFPV-IB is 14.6%, 36.4%, and 2.29% for $T = 3$ s, 4 s, and 5 s, respectively.

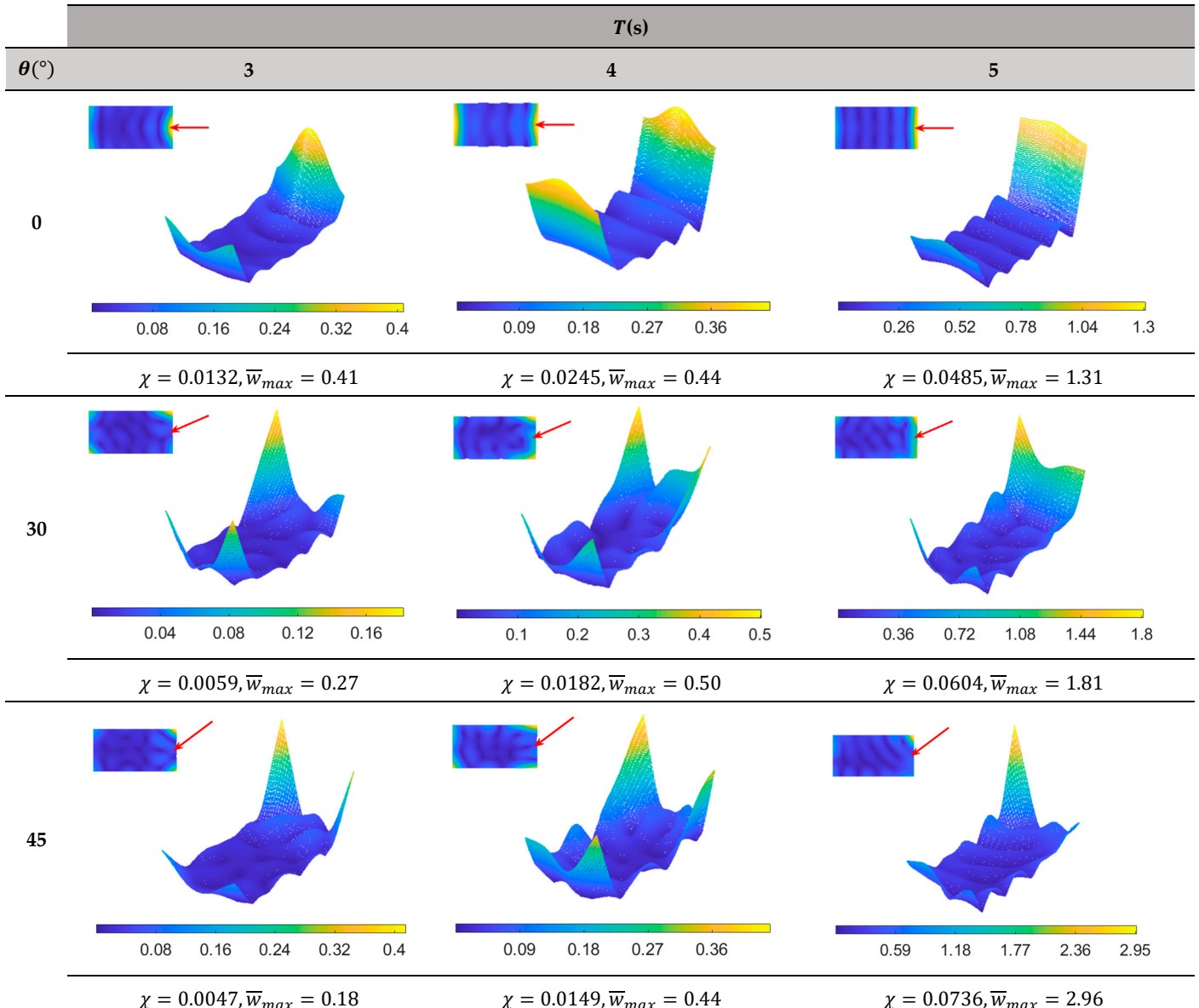

**Figure 9.** Hydroelastic response of OFPV-IC under various wave periods and wave directions. Wave direction is indicated by the red arrow. Small contour figure represents plan view and large contour figure represents isometric view of OFPV-IC.

### 6.1.2. Effect of Layout Configuration

The hydroelastic response of the OFPV could be further mitigated by using modular units with a shorter length to increase the OFPV's overall structural stiffness. Here, modular units with lengths of $l_x = 1.0$ m and $l_y = 0.5$ m are used to form OFPV-I without altering the overall footprint occupied by the OFPV given in Table 1, i.e., when $l_x = l_y = 1.0$ m is used and the overall footprint is kept at 10,000 m². The longitudinal stiffness, i.e., flexural stiffness in the $x$ direction increases when modular units with a shorter $l_y$ are used to form OFPV-I′. A prime ( ′ ) is used to differentiate the OFPV-I given in Table 1 from its counterpart with a higher longitudinal rigidity, i.e., OFPV-I′, OFPV-IA′, OFPV-IB′, and OFPV-IC′ are used for the OFPVs with increased longitudinal stiffnesses. With an increasing longitudinal stiffness, the natural frequencies of OFPV-1A′, OFPV-IB′, and OFPV-IC′ reduce, as shown in Table 6. Note that the (number of rows) × (number of columns) in the grid layout for OFPV-IA′, OFPV-IB′, and OFPV-IC′ are 112 × 73, 100 × 81, and 80 × 102,

respectively, representing an increase in the number of rows when compared to OFPV-IA, OFPV-IB, and OFPV-IC as presented in Figure 3a–c, respectively.

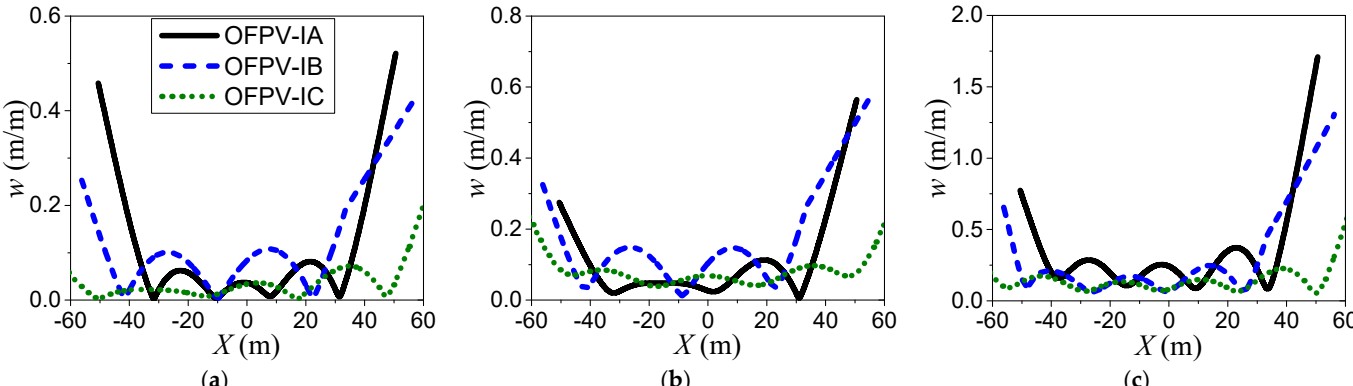

(**a**)  (**b**)  (**c**)

**Figure 10.** Elevated view of hydroelastic response along centreline of OFPV-IA, OFPV-IB, and OFPV-IC under (**a**) $T = 3$ s, (**b**) $T = 4$ s, and (**c**) $T = 5$ s. Incoming waves travel from right to left.

**Table 6.** Natural frequency for OFPV-IA, OFPV-IB, and OFPV-IC ($h = 0.2$ m).

|  | OFPV-IA | | OFPV-IB | | OFPV-IC | |
|---|---|---|---|---|---|---|
| $\bar{h}$ | 0.0146 | | 0.0050 | | $1.4630 \times 10^{-3}$ | |
| Mode | $f_n$(hz) | | $f_n$(hz) | | $f_n$(hz) | |
|  | OFPV-IA' | OFPV-IA | OFPV-IB' | OFPV-IB | OFPV-IC' | OFPV-IC |
| 1 (RBM *) | 0.0000 | 0.0000 | 0.0000 | 0.0000 | 0.0000 | 0.0000 |
| 2 (RBM *) | 0.0000 | 0.0000 | 0.0000 | 0.0000 | 0.0000 | 0.0000 |
| 3 (RBM *) | 0.0000 | 0.0000 | 0.0000 | 0.0000 | 0.0000 | 0.0000 |
| 4 | 0.0080 | 0.0101 | 0.0067 | 0.0098 | 0.0081 | 0.0062 |
| 5 | 0.0095 | 0.0120 | 0.0082 | 0.0102 | 0.0084 | 0.0101 |
| 6 | 0.0131 | 0.0122 | 0.0147 | 0.0153 | 0.0116 | 0.0171 |
| 7 | 0.0181 | 0.0232 | 0.0176 | 0.0226 | 0.0183 | 0.0211 |
| 8 | 0.0205 | 0.0235 | 0.0184 | 0.0255 | 0.0200 | 0.0240 |
| 9 | 0.0288 | 0.0334 | 0.0220 | 0.0271 | 0.0231 | 0.0314 |
| 10 | 0.0314 | 0.0334 | 0.0305 | 0.0407 | 0.0319 | 0.0335 |
| 11 | 0.0322 | 0.0434 | 0.0361 | 0.0422 | 0.0335 | 0.0346 |

Figure 11 shows the hydroelastic response along the centreline of OFPV-I' and OFPV-I under $T = 5$ s and headsea direction ($\theta = 0°$). This reduction in the hydroelastic response is profound for OFPV-IA', as the maximum response is reduced by half when compared to OFPV-IA. Although there is little reduction in the maximum responses for OFPV-IB' and OFPV-IC', they have less elastic deformation due to the greater modular units in the $y$ direction, thus increasing the longitudinal rigidity of the OFPV. As a result, the overall hydroelastic response of OFPV-I' is smaller when compared to OFPV-I. The increase in rows in the grid layout also results in an increase in the mass of the OFPV, thus contributing to the reduction in hydroelastic response.

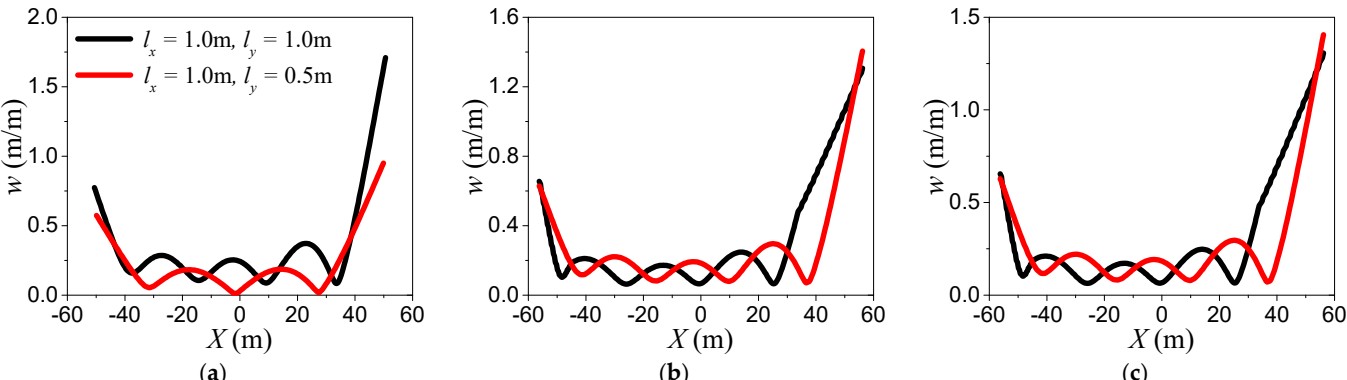

**Figure 11.** Elevated view of hydroelastic response along centreline of (**a**) OFPV-IA', (**b**) OFPV-IB', and (**c**) OFPV-1C', under $T = 5$s and headsea direction. Incoming waves travel from right to left.

### 6.1.3. Compliances

To quantify the deformation of the OFPV, the parameter compliance $\chi$ is introduced [46,47]:

$$\chi = \int_{S_{HB}} w \cdot dS_{HB} \tag{34}$$

The compliance $\chi$ in (34) represents the total volume under the hydroelastic response $w$, where a higher $\chi$ indicates a higher overall response of the OFPV, and vice versa. The $\chi$ values for OFPV-IA, OFPV-IB, and OFPV-IC are presented in Figure 12, which shows that OFPV-IC has the smallest $\chi$ values among the three configurations considered. Thus, this implies that the long-ish layout configuration (small $\alpha$) for the OFPV is desirable, as it has a smaller hydroelastic response.

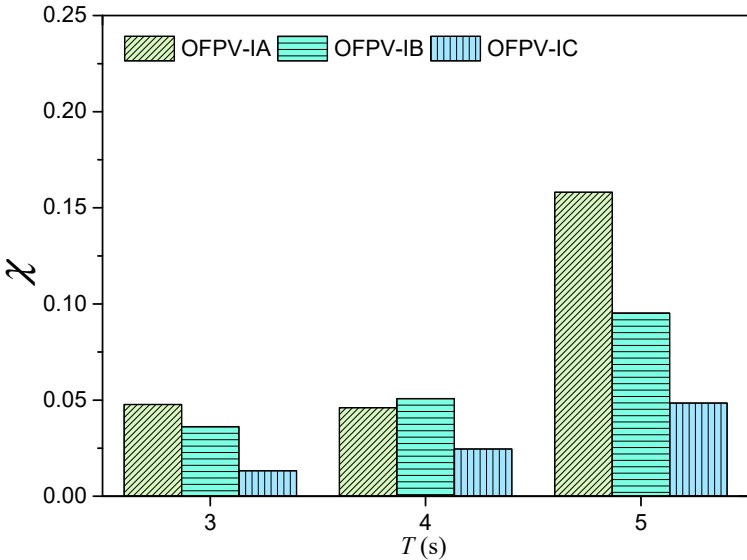

**Figure 12.** Compliance $\chi$ for OFPV-IA, OFPV-IB, and OFPV-IC.

In the next section, considerations are given by splitting the OFPV-I into several separate layouts as presented for each OFPV.

In Figure 3f is termed OFPV-II, similarly, the hydroelastic response will be investigated and the compliances $\chi$ for each OFPV-II will be compared with their counterparts for OFPV-I.

### 6.2. OFPV-II

In this section, details are given for OFPV-IIA, OFPV-IIB, and OFPV-IIC.

Figure 3 and Table 1 are considered in this section. Due to space constraints, only the headsea condition, where the compliances $\chi$ for OFPV-I are the highest (presented in Figure 12), is considered. The OFPVs are separated by spacings of $s_p = 1$ m, 5 m, and 10 m to study the effect of gap spacing on the hydroelastic response. Comparisons of the compliance $\chi$ between OFPV-II and OFPV-I under $T = 3$ s, 4 s, and 5 s are presented in Figure 13. Figure 13 shows that separating the OFPV-I into smaller modules does not result in a reduction in the overall hydroelastic response. The $\chi$ value for OFPV-I (floating farm interconnected in one whole piece, shown by the black bar in Figure 13) has a significantly lower response under wave action as compared to the separated counterparts. In general, when comparing the $\chi$ for the three OFPV-II configurations, separating the OFPV into more modules, i.e., OFPV-IIA, results in a higher compliance due to the greater interference effect between the four separated OFPVs. In addition, as the separated farm in OFPV-II has smaller mass, this results in an increase in the hydroelastic response as compared to OFPV-I. The comparisons also show that the hydroelastic response for the square OFPV-IIA increases with the increase in gap spacing $s_p$, whereas the response for the long-ish OFPV-II, i.e., OFPV-IIC, reduces with the increase in $s_p$.

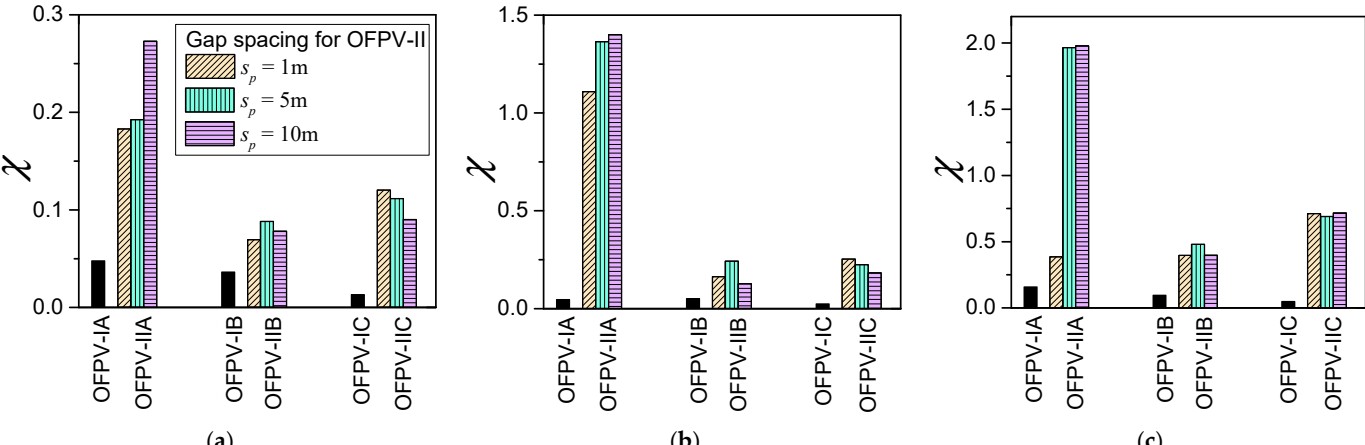

**Figure 13.** Comparison of compliance $\chi$ between OFPV-I and OFPV-II under (**a**) $T = 3$ s, (**b**) $T = 4$ s, and (**c**) $T = 5$ s.

The deflection contours for OFPV-IIA, OFPV-IIB and OFPV-IIC are presented in Figures 14–16 for $s_p = 1$m, 5m and 10m, respectively. The OFPVs that are first hit by the incident waves have higher responses as compared to their counterparts located on the leeward side of the layouts. The $\chi$ for each separate OFPV labelled in Figures 14–16 reveals that a windward-positioned floating structure could serve as a buffer unit that attenuates the incoming waves before hitting the leeward-positioned OFPV. This suggests that placing a floating breakwater [48,49] at the windward side of the layout might be effective in mitigating the hydroelastic response of the OFPV. Alternatively, the OFPV platform located at the windward side of the layout may be stiffened up such as by using material with a higher Young's modulus $E$ or using modular units with greater depth $h$. Other means such as altering the layout configuration of the OFPV [50] or using an articulated plate anti-motion device [51,52] could be adopted to mitigate the hydroelastic response of the OFPV. It is also interesting to note that the $\chi$ values for OFPV-I are still lower compared to their separated counterparts considered in OFPV-II. Therefore, this finding suggests that OFPVs should be interconnected in one whole piece, if possible, to reduce the hydroelastic response. It is also important to consider the stress-resultants for a connector design that integrates the modular units together for the future work.

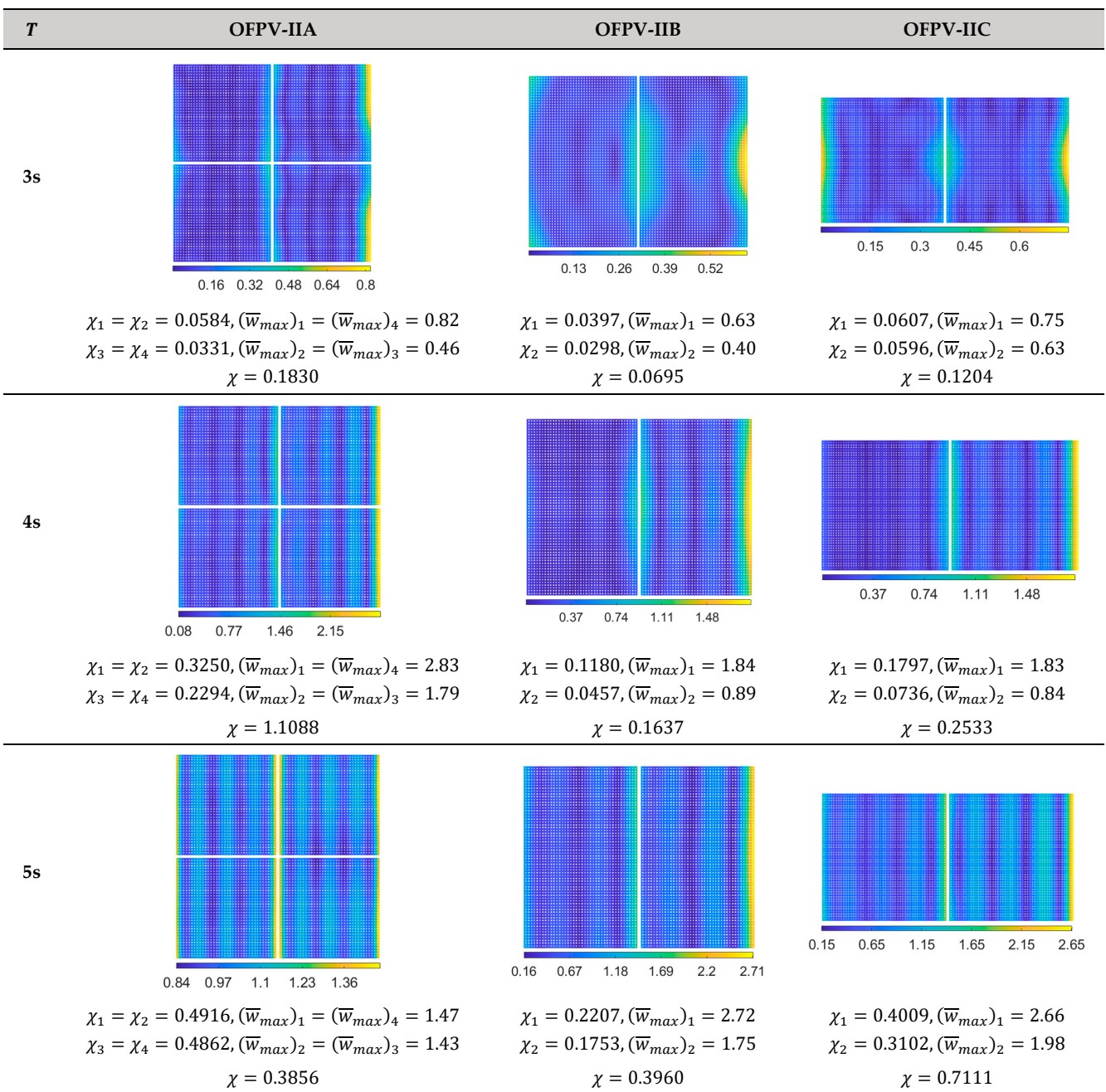

**Figure 14.** Plan view of hydroelastic response for OFPV-II with $s_p = 1$ m. Regular wave, headsea condition. Waves travel from right to left. Note: The subscript in $\chi$ and $\overline{w}_{max}$ represents the module number in for each OFPV.

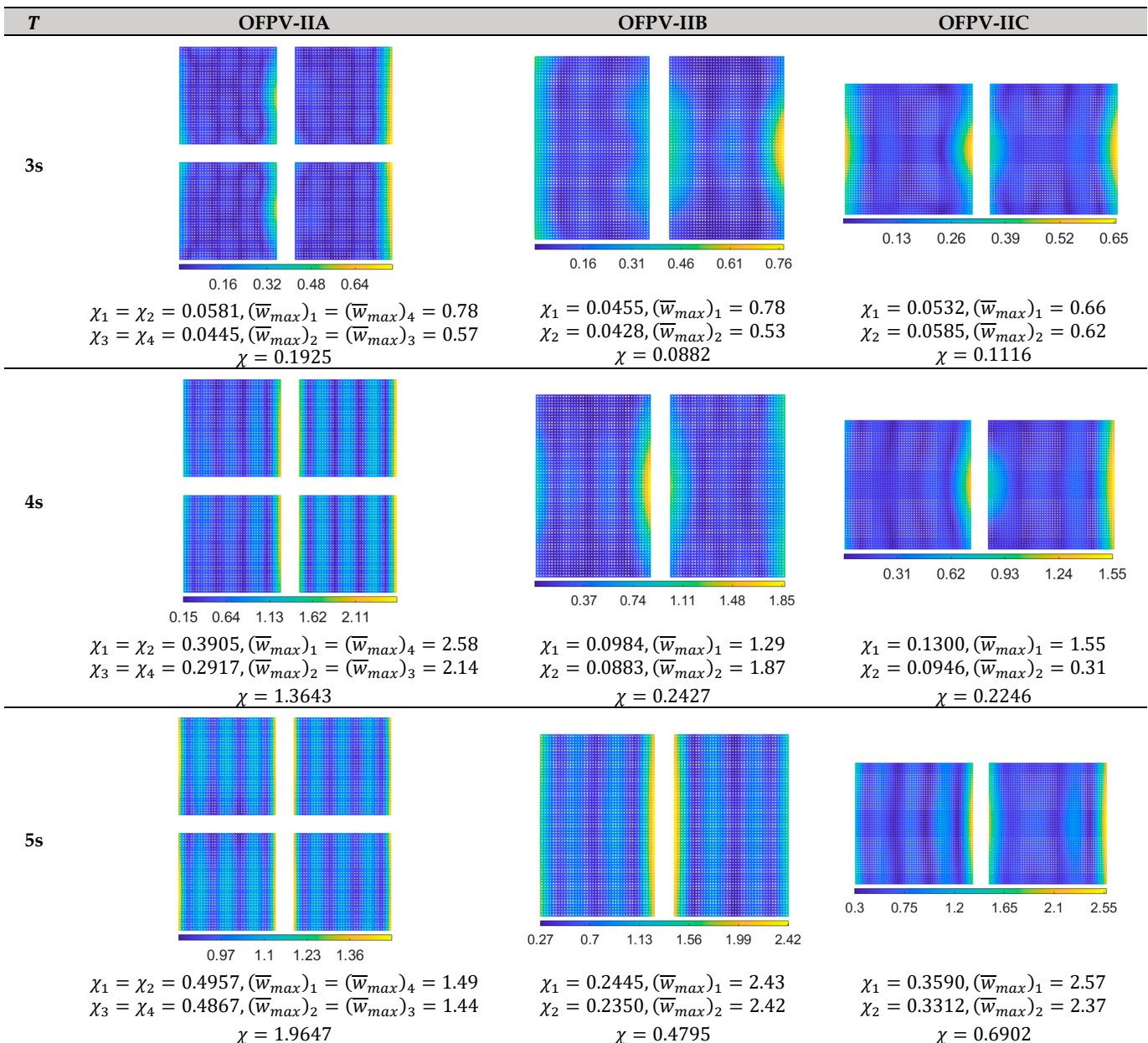

**Figure 15.** Plan view of hydroelastic response for OFPV-II with $s_p = 5$ m. Regular wave, headsea condition. Waves travel from right to left. Note: The subscript in $\chi$ and $\overline{w}_{max}$ represents the module number in for each OFPV.

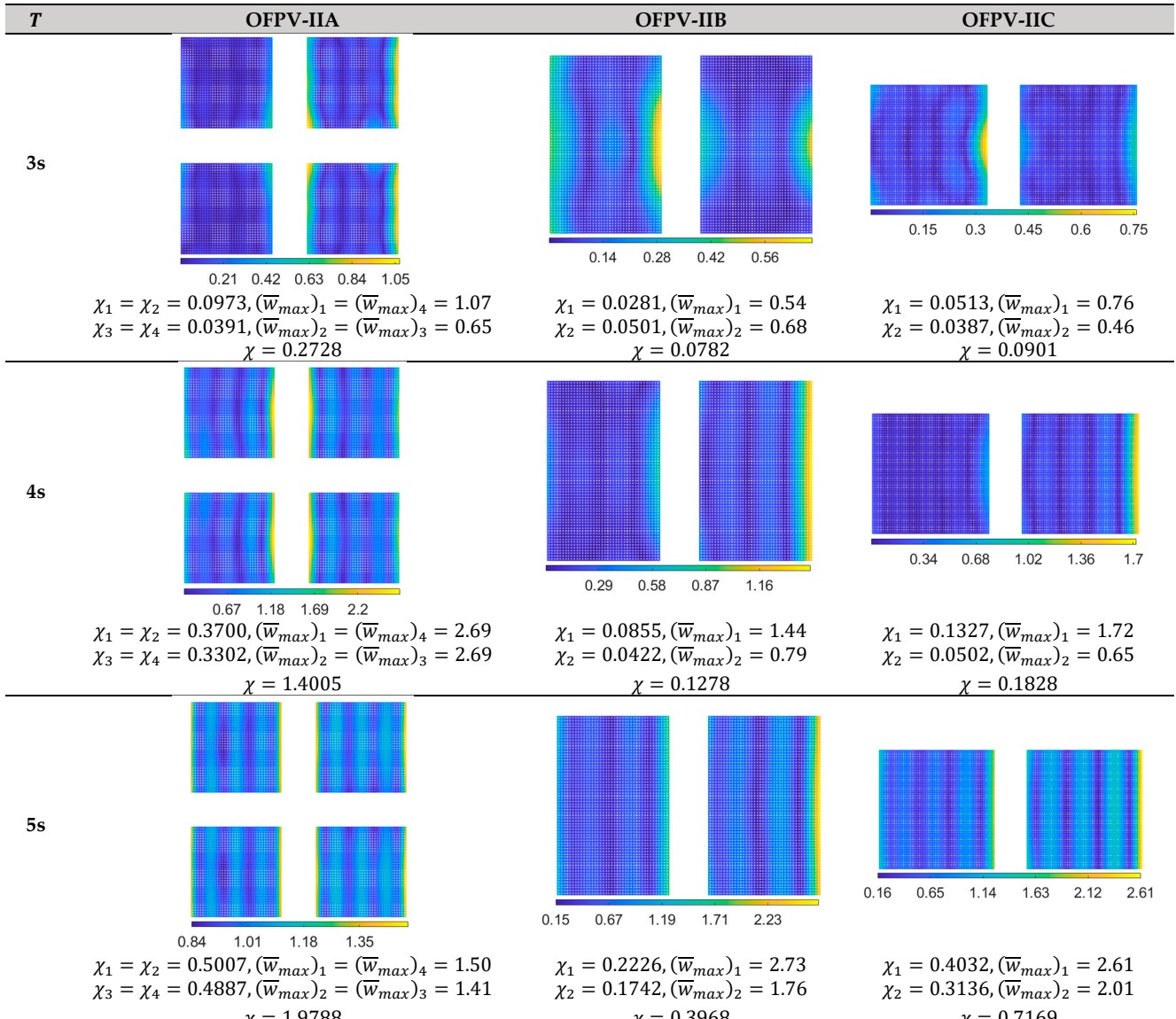

**Figure 16.** Plan view of hydroelastic response for OFPV-II with $s_p = 10$ m. Regular wave, headsea condition. Waves travel from right to left. Note: The subscript in $\chi$ and $\overline{w}_{max}$ represents the module number in for each OFPV.

## 7. Conclusions

A three-dimensional hydroelastic analysis of OFPVs was presented, where the floating structure was assumed to be a mat-like VLFS in grid configuration that could be modelled using the Kirchhoff–Love thin plate theory, whereas the water was assumed to be an ideal fluid modelled using the potential wave theory. The hybrid boundary element-finite element method was used to solve the fluid-structure interaction problem. The free vibration analysis was first carried out by solving the eigenvalue problem, where the natural frequencies (eigenvalue) and vibration modes (eigenvectors) were verified with their counterparts obtained from the finite element software ABAQUS. The verification showed good agreement in the natural frequencies and modes between the present model and those obtained from ABAQUS. This paper then proceeded to study the hydroelastic response of the OFPVs, where two case studies were carried out, i.e., OFPV-I (OFPV in one whole piece) and OFPV-II (OFPV in separate modules). It is important to note that the present method is limited to OFPVs with a small thickness-to-structural length ratio due to the limitation of the Kirchhoff–Love thin plate theory. A higher order plate theory such as

the first order or third order shear deformation plate theory could be used to model OFPVs with a larger thickness. As a conclusion, the following findings were obtained:

- The hydroelastic response and compliance for OFPV-I showed that the elastic deformation of the OFPV increases with a reduction in wave periods;
- OFPVs with a smaller aspect ratio (long-ish structure) have greater elastic deformation but deflect in smaller magnitudes compared to the square OFPV-IA (aspect ratio $\alpha = 1.0$);
- The hydroelastic response under headsea conditions could be reduced by increasing the longitudinal stiffness of the OFPV. This can be done by reducing the spacing between the longitudinal modules in the OFPV;
- By splitting the OFPV into smaller OFPVs, i.e., OFPV-II, the compliance $\chi$ values implied that the hydroelastic response was smaller for OFPV-I when it wasconstructed in one piece;
- The gap spacing between the separated farms in OFPV-II showed a profound difference in the hydroelastic response due to the interference effect between the separated farms, and OFPV-IIA, which comprises four separated farms, has a higher $\chi$ compared to OFPV-IIB and OFPV-IIC, both having two separated farms;
- The plot contours of the hydroelastic deflection showed that the farms located on the leeward side have smaller $\chi$ values compared to their counterparts located on the windward side.

In conclusion, this paper presents a numerical framework for computing the hydroelasticity of OFPV farms and the results presented here provide insight into the preferable layout configuration for OFPVs subject to wave action.

**Funding:** The author would like to express his gratitude to the resources provided by MOE, Grant Number R-MOE-E103-F010 and F-MOE-A204-G005.

**Data Availability Statement:** Not applicable.

**Conflicts of Interest:** The authors declare no conflict of interest.

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
