# Peer review of "Three-Dimensional Hydroelasticity of Multi-Connected Modular Offshore Floating Solar Photovoltaic Farm"

_jmse, doi:10.3390/jmse11101968_

Round 1
Reviewer 1 Report
(1) Please specify the material (by the density and the modulus of elasticity I guess it is HPDE).
(2) Line 161, I assume the unit is MPa (as specified in line 174 below Table 1) and not GPa. Please check and correct.
(3) The density 960 kg/m3 does not provide much residual flotation for the weight of the solar panels. Please address this issue.
(4) I assume that the solar panels are brittle, and cannot be deformed as the carrying structure, which this research analyses, so the panels should be small, and connected to the structures on dampers. Please address this issue.
(5) What is d in line 149? Is it h? I did not find a definition of d.
(6) Considering the bars of the grid structures, the cross section dimension: b = 0.4m and h = 0.2m or 0.6m are more like Beam theory and not Plate theory. Please address this point.
(7) In view of Figure 3 and equations (3) and (5), it is not clear to me wheather you analyses the actual geometry of the grid or an equivalent mat with no square spaces. Do you apply the free surface B.C. to the water in the spaces? Does the BEM and the FEM mesh the actual geometry? Pleas add a Figure that demonstrate the mesh for the BEM and the FEM.
(8) Line 215, “The boundary conditions at the free edges of the floating plate are, ” is not completed.
(9) Line 267, to obtain the vertical loads, the pressure should by the horizontal wetted surface and not the vertical (or did you mean that the Normal is vertical?). Please check and clarify.
(10) The presented results are for the deformations only. For design it is important to consider also moments and stresses and to assess the structure durability to withstand storms, or the allowable wave conditions.
Reviewer 2 Report
This paper investigates the hydroelastic responses of offshore floating solar photovoltaic farms to minimize structural motion based on free edge conditions. The mathematical equation involved in the formulation are well supported by the references. For solution, the hybrid boundary element-finite element method was used and the verification of the present results was performed with results obtained from ABAQUS. The present work has a potential to be published in JMSE in the specific field of Hydroelasticity. However, the present paper needs to be a major revision for wider readability and improve the content of the manuscript. My suggestions and comments are cited below:
1. Introduction is missing some recent potential works conducted by other researchers that are very relevant to the present study. For example:
https://doi.org/10.1016/j.oceaneng.2022.110785
https://doi.org/10.1016/j.jfluidstructs.2022.103588
https://doi.org/10.3390/jmse10091205
2. Figure 1 should be re-sketched in three-dimensional form for better clarity of the model.
3. Equation 20 should be supported by the references.
4. In Figure 9, the difference of percentage between the comparisons should be discussed.
5. The detail of the computational cost of the numerical results should be mentioned.
6. Conclusion Section: The discussion of limitations and further research direction of the present work need to be discussed. Further, using bullet points, the significant concluding remarks need to be reorganized instead of essay form.
Minor editing of English language is required.
Reviewer 3 Report
(Table.1)
How did the regular wave conditions get set up? For now, it looks like the ordinary regular waves were set without much thought. They are far enough away from the resonant periods in Tables 2-6 that the considered regular wave conditions are unlikely to have much engineering significance.
(Chapter. 4)
None of the mathematical formulas described are new to this paper, so please just cite the appropriate papers.
(Fig. 9-10)
Almost all structures have a large deformation at x=50 meters. This is different from when x=-50m, so I think the boundary conditions are wrong in the hydroelastic analysis, and the elastic behavior needs to be redone. The current results do not make sense physically.
(Overall)
The paper appears to be an interpretive look at how different arrangements and water depths might affect things. Could you please organize your results so that we can see them more clearly?
Also, it would be nice to see more detailed modeling of the hinges in the next study, as well as the durability issues caused by blue water.
Professional English correcting service needs to be served.
Round 2
Reviewer 1 Report
No Comments.
Reviewer 2 Report
The author revised the manuscript based on my previous comments. However, one of my suggestions did not consider at all to the present study. I'm sure the aspects of the present paper is broad and believe that the author must consider at least one reference to the present study (https://doi.org/10.1016/j.oceaneng.2022.110785) as it involves numerical, segmented structures (multi-mudulo) and also gives hydroelastic effect.
Still, minor editing of English language is needed.
Reviewer 3 Report
This revised article seems to be acceptable as a journal paper of JMSE.
Acceptable